# A dendritic hexamer acceptor enables 19.4% efficiency with exceptional stability in organic solar cells

Tao Jia[1,2,7], Tao Lin[1,7], Yang Yang[1,7], Lunbi Wu[1], Huimin Cai[1], Zesheng Zhang[3], Kangfeng Lin[1], Yulong Hai[4], Yongmin Luo[4], Ruijie Ma [5] ✉, Yao Li[4], Top Archie Dela Peña [4], Sha Liu[6], Jie Zhang[3], Chunchen Liu[3], Junwu Chen [3], Jiaying Wu [4] ✉, Shengjian Liu [2] ✉ & Fei Huang [3]

To achieve the commercialization of organic solar cells (OSCs), it is crucial not only to enhance power conversion efficiency (PCE) but also to improve device stability through rational molecular design. Recently emerging giant molecular acceptor (GMA) materials offer various advantages, such as precise chemical structure, high molecular weight (beneficial to film stability under several external stress), and impressive device efficiency, making them a promising candidate. Here, we report a dendritic hexamer acceptor developed through a branch-connecting strategy, which overcomes the molecular weight bottleneck of GMAs and achieves a high production yield over 58%. The dendritic acceptor Six-IC exhibits modulated crystallinity and miscibility with the donor, thus better morphology performance compared to its monomer, DTC8. Its charge transport ability is further enhanced by additional channels between the armed units. Consequently, the binary OSCs based on D18:Six-IC achieves a cutting-edge efficiency of 19.4% for high-molecular weight acceptor based systems, as well as decent device stability and film ductility. This work reports high-performance OSCs based on dendritic molecule acceptor with a molecular weight exceeding 10000 g/mol and shares the understanding for designing comprehensively high-performing acceptor materials.

Organic solar cells (OSCs) with >20% reported power conversion efficiencies (PCEs) requires more comprehensive progress on other figure-of-merits such as device stability and cost effectiveness[1–16]. Compared to traditional polymer donor and small molecule acceptor combinations that yield high efficiencies[17–20], all-polymer systems, with their inherently better mechanical and thermal stability due to the increased molecular weight of the acceptor material, are considered more promising if their device efficiencies are equally high—a potential that has been proven practical benefiting from the emergence of polymerized small molecule acceptors (PSMAs) in recent years[21–32].

[1]School of Optoelectronic Engineering, Guangdong Polytechnic Normal University, Guangzhou, China. [2]School of Chemistry, Guangzhou Key Laboratory of Materials for Energy Conversion and Storage, Key Laboratory of Electronic Chemicals for Integrated Circuit Packaging, South China Normal University (SCNU), Guangzhou, China. [3]Institute of Polymer Optoelectronic Materials and Devices, State Key Laboratory of Luminescent Materials and Devices, South China University of Technology, Guangzhou, China. [4]Advanced Materials Thrust, Function Hub, The Hong Kong University of Science and Technology (Guangzhou), Nansha, Guangzhou, China. [5]Department of Electrical and Electronic Engineering, Research Institute for Smart Energy (RISE), Photonic Research Institute (PRI), The Hong Kong Polytechnic University, Hong Kong, China. [6]Dongguan Key Laboratory of Interdisciplinary Science for Advanced Materials and Large-Scale Scientific Facilities, School of Physical Sciences, Great Bay University, Dongguan, Guangdong, China. [7]These authors contributed equally: Tao Jia, Tao Lin, Yang Yang. ✉e-mail: ruijie.ma@polyu.edu.hk; jiayingwu@ust.hk; shengjian.liu@m.scnu.edu.cn

However, PSMAs face several notable drawbacks: (1) efficient PSMAs typically require isomeric purification of IC terminals and a lengthy synthetic pathway, adding considerable complexity to their target production; (2) due to the lack of a defined structure, PSMAs experience significant batch-to-batch variability[33–35], creating challenges for commercial production. Therefore, developing acceptor materials with high yields, reproducibility, and substantial molecular weight presents a critical opportunity for advancing OSCs toward future commercialization.

In recent years, a concept of giant molecular acceptors (GMAs), derived from polymerized small molecule acceptors, has been proposed. This innovation offers a promising opportunity to achieve both improved PCE and enhanced stability (light, thermal, and mechanical) in OSCs[36–52]. However, this strategy introduces several challenges: (i) GMAs struggle to achieve decent efficiency when their molecular weight exceeds 10,000 g/mol due to poor molecular packing and charge transport; (ii) Precisely synthesizing GMAs, particularly for multimers larger than trimers, results in significant synthetic difficulties and low yields, which severely limits cost-effectiveness. In light of these challenges, there is a pressing need for the designs of high molecular weight, easily synthesized GMAs that can achieve eminent PCEs.

In this report, we apply a branched molecule design strategy, successfully producing a dendritic hexamer acceptor called Six-IC, enabled by very simple synthetic routes with satisfactory yields. Six-IC possesses a precise chemical structure and an high molecular weight exceeding 10,000 g/mol. Compared to its monomer counterpart, DTC8 (PCE = 17.6%), Six-IC achieves a 19.4% efficiency when paired with D18, owing to enhanced charge generation, multichannel charge transport, and reduced exciton recombination. Notably, very few GMAs have been reported to achieve over 19% efficiency, especially for such a high molecular weight. MD simulations suggest that Six-IC possesses additional charge transport channels and tighter packing within its arms. Despite its branching structure, no significant reduction in crystallinity was observed in the blends. Besides, Six-IC's glass transition temperature ($T_g$) was significantly promoted in comparison with that of DTC8. Reasonably, the thermal stability of the device and the stretchability of the active layer are improved in the Six-IC-based system. Six-IC-based devices retain over 94.2% and 98.0% of their PCE after 1200 h of thermal annealing at 85 °C and storage, respectively, significantly outperforming the stability of DTC8-based devices. The blend film's ductility is also enhanced for D18:Six-IC, likely due to the increased molecular weight and potentially strengthened homologous and heterologous molecular interactions. Overall, this work presents a high-efficiency hexamer acceptor, offering enhanced thermal stability and ductility without sacrificing efficiency.

## Results
### Polymer design, synthesis, and characterization
The synthesis route for realizing branched hexamer Six-IC is illustrated in Fig. 1. By modifying the synthesis procedure reported[53], the commercially available compound 1 can be converted into bromide alkyl functionalized precursor 2 in relatively high yield of 82%. Then the hexamer's skeleton Six-H was obtained by the nucleophilic reaction of cyclotriveratrylene (CVT)-derived core and armed compound 2 in 86% yield. The effectiveness of the nucleophilic reaction between brominated alkanes and phenolic hydroxyl groups, combined with the non-conjugated CVT core, ensures that the chemical reactivity of each arm remains independent. These factors collectively contribute to the high yield of the key intermediate, Six-H. Different from many GMA structures obtained using aromatic ring linkers, the key six-armed intermediate, Six-H, can be achieved with a notable yield of over 70% for two-step reaction. The transformation of Six-H into the dialdehyde compound, Six-CHO, can be accomplished through a Vilsmeier–Haack reaction with high yield of 92%. Subsequently, a pyridine-catalyzed

Knoevenagel condensation can be employed to convert Six-CHO into the desired target dendritic molecule acceptor Six-IC in 90% yield. The corresponding structural characterizations involving [1]H NMR, [13]C NMR, mass spectra, and high-temperature GPC data were summarized in Supplementary Figs. 1–10. It's noteworthy that, owing to the comparatively high yields achieved in intermediate reactions, the overall yields of Six-IC reached up to an impressive 58%. This value represents a relatively high yield for synthesizing high-molecular-weight GMAs (Supplementary Table 1), underscoring the potential of the branched molecule design strategy for acceptor synthesis and future applications.

### Optical, electronic, and thermal properties
The chemical structures of D18, Six-IC, and the control SMA, DTC8, are shown in Fig. 2a. As illustrated in Fig. 2b, in addition to the peak at 732 nm, an additional peak appears at a blue-shifted position of 689 nm, which is mainly attributed to the H-aggregates of the DTC8 moiety. Simultaneously, the solution absorption onset exhibits a significant red shift, confirming that the branched Six-IC with a flexible spacer ensures distinctive packing of the SMA arms. When it comes to film absorption, both DTC8, and Six-IC exhibit significant redshifts; however, Six-IC shows a 46 nm blue-shifted 0-0 absorption peak compared to DTC8 (Fig. 2c). This indicates that Six-IC and DTC8 follow different aggregation motifs in liquid and solid states, and dendritic structure can effectively regulates molecular aggregation[54,55]. The relevant detailed data are summarized in Table 1. The molecular energy levels of the active layer materials were estimated using ultraviolet photoelectron spectroscopy (UPS) and inverse photoemission spectroscopy (IPES) measurements (Supplementary Fig. 11). As shown in Fig. 2d and Table 1, the HOMO/LUMO energy levels obtained by UPS/IPES were −5.89/−4.05 eV for DTC8, and −5.84/−3.92 eV for Six-IC, respectively. These results indicate a simultaneous increase in both the HOMO and LUMO energy levels for the dendritic molecule Six-IC. The elevated LUMO energy level of Six-IC is favorable for achieving a higher open-circuit voltage. Thermogravimetric analysis results indicate that the dendritic structure design does not raise any stability concerns (Supplementary Fig. 12). The $T_g$ values of Six-IC and DTC8 were first determined using the UV-vis deviation metric (DMT) method (Fig. 2e, f) to investigate the onset of molecular diffusion and relaxation[56]. Six-IC exhibits a higher $T_g$ of 182 °C compared to DTC8, which has a $T_g$ of 97 °C. As shown in Fig. 2g, DTC8 exhibited a cold crystallization temperature ($T_c$) of 150 °C, while Six-IC showed no noticeable $T_c$. Additionally, Six-IC displayed a much higher melting temperature ($T_m$) at 254 °C compared to DTC8, which melted at 203 °C. The melting enthalpy ($\Delta H_m$) of Six-IC decreased significantly, from 23.85 J/g to 9.47 J/g. These findings suggest that our developed dendritic acceptors can effectively suppress crystallization, potentially limiting the diffusion of SMA in polymer blends[57].

To understand how branched structure engineering affects the intramolecular packing of SMA subunits in Six-IC, the intermolecular packing of DTC8 and the electronic characteristics of Six-IC were analyzed using density functional theory (DFT) calculations. The source data are provided in Supplementary Data 1 and 2. Considering that the electronic structure is not significantly affected by the number of SMA units, and to balance calculation time with simulation accuracy, a simplified Six-IC model containing only two SMA subunits was used for the calculations. As depicted in Supplementary Fig. 13a–c, both single DTC8 and SMA subunits within Six-IC display planar molecular geometries. The dipole moment of the branched Six-IC was studied, revealing a direction from the side-chain SMA subunits toward the cyclotriveratrylene core, with a value of 6.98 Debye (Supplementary Fig. 13c). As shown in Fig. 3a, Six-IC exhibited a neutral electrostatic potential (ESP) on the cyclotriveratrylene core and a negative ESP distribution on the SMA arms. The HOMO and LUMO energy orbitals are primarily located on the SMA units, as illustrated in Fig. 3b, which

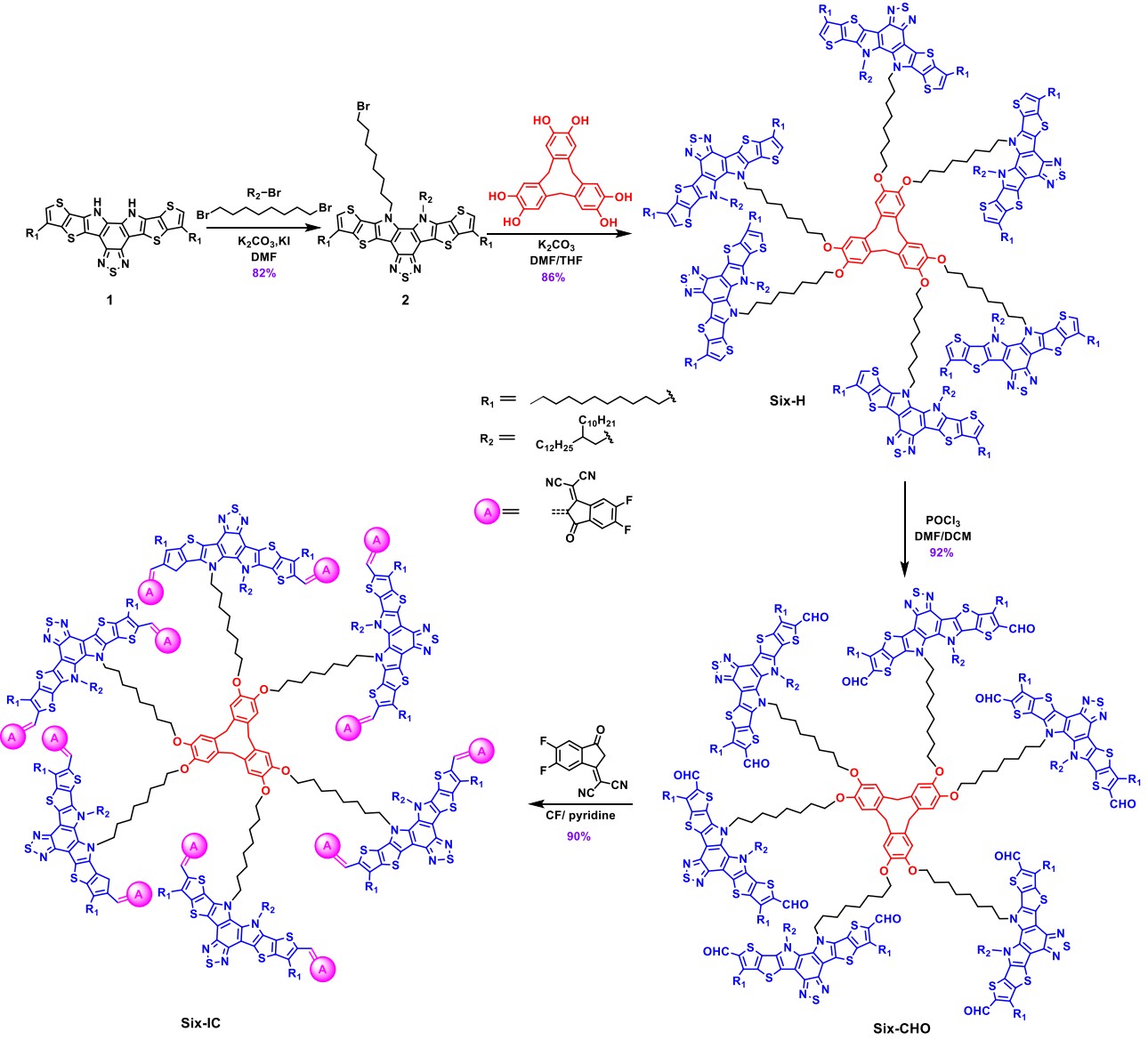

**Fig. 1 | Synthetic routes.** Synthetic routes of the intermediates and Six-IC.

facilitates charge transfer between SMA molecules. Supplementary Fig. 14 illustrates the electron and hole distributions for DTC8 and Six-IC, with blue representing electrons and red representing holes. A small hole distribution in the central core suggests the potential for the cyclotriveratrylene unit to transfer electrons to the SMA subunits. In Fig. 3c, red indicates a decrease in electron density, while blue indicates an increase. It is evident that the cyclotriveratrylene core loses electrons while the SMA arms gain electrons, confirming the presence of charge transfer from the central core to the side-chain SMA unit. It is important to note that the charge transfer from the central core to the side-chain SMA units may provide an alternative charge transport channel, potentially reducing charge recombination losses. The intermolecular packing properties of DTC8 and Six-IC were further investigated through molecular simulation (MD) (Fig. 3d). The source data are demonstrated by Supplementary Data 3−6. Due to the cup-shaped conformation of cyclotriveratrylene, the SMA subunits of Six-IC remain on the same side. The SMA arms of Six-IC exhibits more ordered intermolecular packing (3.03−3.56 Å), whereas DTC8 shows relatively chaotic packing (3.47−4.35 Å). These results were primarily attributed to the combination of the branched structure and flexible spacer linkage, which allows the SMA subunits to stack without

obstruction. The more compact packing of Six-IC can lead to more effective charge transport[58]. Theoretically, we demonstrate that the branched structure can create specific spatial arrangements that allow for more efficient packing, contrary to the intuitive assumption that it might hinder molecular packing.

## Photovoltaic properties

The device performance is assessed by fabricating a series of solar cells with conventional structures based on ITO/PEDOT:PSS/active layer/PNDIT-F3N/Ag. The optimal current density versus voltage (J-V) characteristics are plotted in Fig. 4a, with extracted parameters put in Table 2. Six-IC-based devices exhibit a significantly higher open-circuit voltage ($V_{OC}$) of 0.92 V and a fill factor (FF) of 78.5%, which fully compensates for the slightly lower short-circuit current density ($J_{SC}$) and ultimately boosts the PCE to 19.4%. To assure the measurement accuracy, the external quantum efficiency (EQE) spectra of optimized devices are presented in Fig. 4b. Accordingly, the integrated current density (also shown in Table 2) values indicate the measurement errors are no more than 5%. Notably, such a high PCE is rarely reported in binary OSCs based on high molecule-weight acceptors, especially considering the unusually high molecular weight of Six-IC. Therefore,

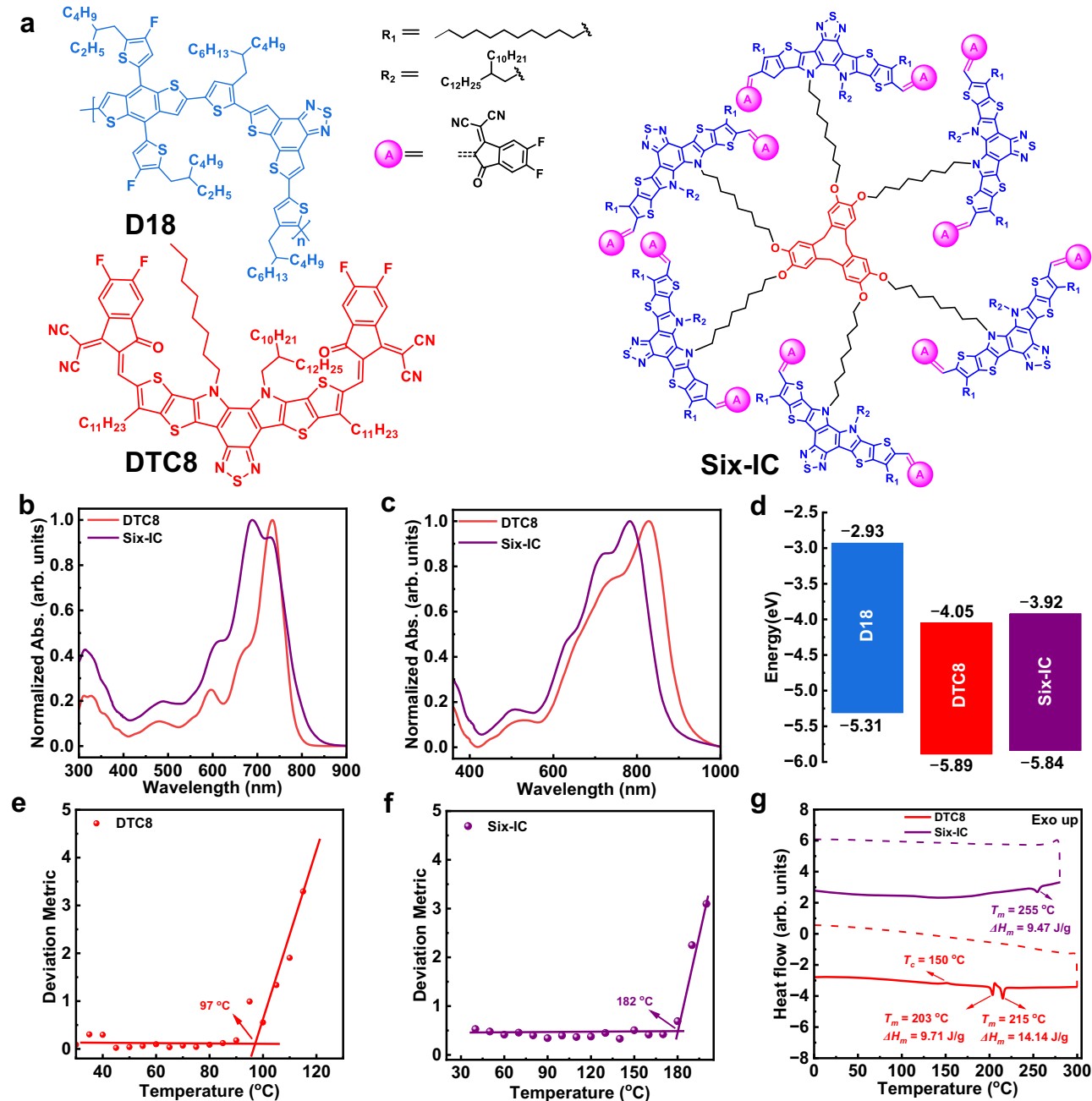

**Fig. 2 | Chemical structures of photoactive materials and the related physical properties. a** Chemical structures of D18, DTC8, and Six-IC. UV-vis absorption spectra of DTC8 and Six-IC in (**b**) solution state and (**c**) films. **d** Energy level distribution of D18, DTC8, and Six-IC. DMT results of (**e**) DTC8 and (**f**) Six-IC. **g** DSC curves of DTC8 and Six-IC (Solid lines and dashed lines represent the heating and cooling processes, respectively).

**Table 1 | The molecular weight, optical properties, energy levels, and thermal properties**

| Acceptor | M [g/mol] | $\lambda_{max}^{sol.}$ [nm] | $\lambda_{max}^{film}$ [nm] | $\lambda_{onset}^{film}$ [nm] | $E_g^{opt\ a}$ [eV] | $E_{HOMO}^{\ b}$ [eV] | $E_{LUMO}^{\ c}$ [eV] | $T_c$ [°C] | $T_m$ [°C] |
|---|---|---|---|---|---|---|---|---|---|
| DTC8 | 1676 | 734 | 829 | 942 | 1.32 | −5.89 | −4.05 | 150 | 203,215 |
| Six-IC | 10,412 | 689,732 | 783 | 906 | 1.37 | −5.84 | −3.92 | – | 254 |

$^a E_g^{opt} = 1240/\lambda_{onset}^{film}$.
$^b$Estimated from UPS measurements.
$^c$Estimated from IPES measurement.

Fig. 4c has been created to highlight our achievement in simultaneously attaining high efficiency and high molecular weight in an acceptor material. Detailed information is provided in Supplementary Table 1.

## Discussion

To better understand the device physics, a set of characterizations is implemented. Firstly, photocurrent density ($J_{ph}$) versus effective voltage ($V_{eff}$) curves were plotted to investigate exciton dissociation and

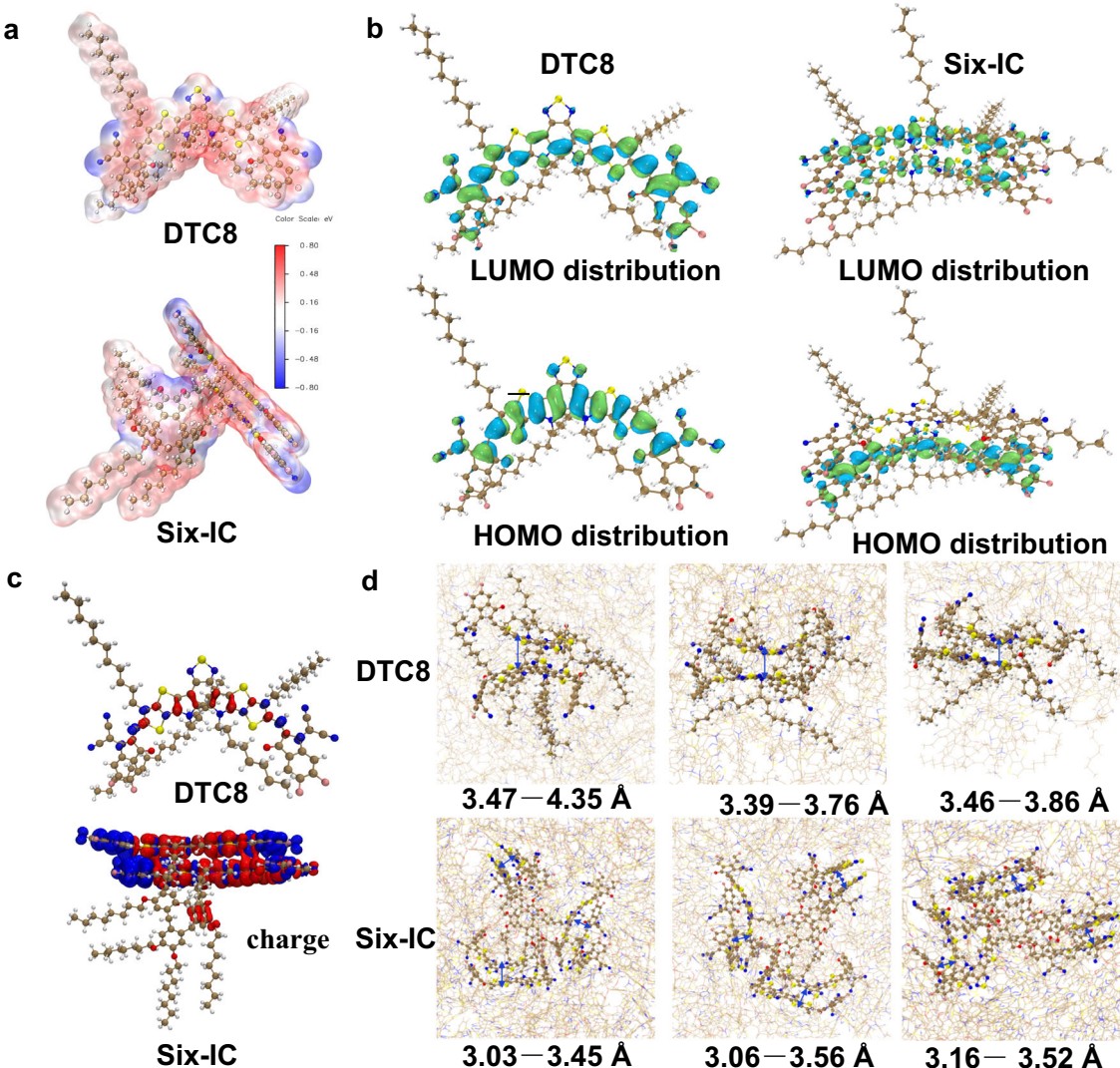

**Fig. 3 | DFT calculation and molecular simulation (MD) results. a** The electrostatic potential (ESP) distributions. **b** The HOMO, LUMO, and energy levels. **c** Charge density differences. **d** Intermolecular interaction and packing patterns of DTC8 and Six-IC from MD simulation.

charge generation in devices (Fig. 4d). Six-IC-based device receives an increase of the $\eta_{diss}$ of 97.8% relative to DTC8-based one (96.7%). These results indicate that exciton dissociation in these devices is highly efficient, and the branched strategy can even slightly enhance the exciton dissociation process. Secondly, charge transport properties of D18:DTC8 and D18:Six-IC were evaluated by the photo-induced carrier extraction with linearly increasing voltage (photo-CELIV) measurement, which can directly measure the majority carrier's mobility in devices. As shown in Fig. 4e, Six-IC-based device exhibited slightly higher charge carrier mobility of $2.18 \times 10^4$ cm$^2$ V$^{-1}$ s$^{-1}$ in comparison with DTC8-based one ($1.83 \times 10^4$ cm$^2$ V$^{-1}$ s$^{-1}$), implying higher carrier mobilities of the bilayer interdiffusion device based on Six-IC[59]. This result can be further verified by the higher SCLC mobilities observed in Six-IC-based films (Supplementary Figs. 15–16). Thirdly, carrier recombination behaviors were examined through charge-extraction (CE) and transient photovoltage (TPV) measurements under varying illumination intensities, along with the trap density of states (DOS) analysis[60]. As shown in Fig. 4f, the Six-IC-based device exhibits longer carrier lifetimes ($\tau$) at equivalent carrier densities ($n$), indicating more efficient exciton dissociation and a lower carrier recombination rate[61]. The nongeminate recombination rate constant ($k_{rec}$) was further calculated using the equation $k_{rec} = 1/[(\lambda + 1)n\tau]$, where $\lambda$ represents the recombination order, to assess the bimolecular recombination

(Supplementary Fig. 17). The Six-IC-based device exhibits a lower $k_{rec}$, primarily due to the multi-armed structure of Six-IC, which increases the heterogeneous molecular contact interface and suppresses bimolecular recombination losses. The trap density of states (DOS) of two devices is analyzed by measuring capacitance–frequency spectra (Fig. 4g). D18:Six-IC device displays a lower energy level of trap states (0.509 eV) and trap DOS ($7.69 \times 10^{16}$ cm$^{-3}$) than D18:DTC8 film (0.519 eV, $9.66 \times 10^{16}$ cm$^{-3}$), which helps minimize recombination losses and contributes to a higher photocurrent[62]. To understand why side-chain branched engineering significantly increases the $V_{OC}$ of binary OSCs, the energy losses were investigated (Supplementary Fig. 18)[63]. As shown in Fig. 4h and Supplementary Table 2, the OSCs based on D18:Six-IC have wide band gaps ($E_g$) of 1.48 eV compared to the D18:DTC8-based one (1.43 eV), implying the branched strategy can broaden the $E_g$. Two systems possess almost identical $\Delta E_1$ and $\Delta E_2$, while Six-IC demonstrated slightly lower non-radiative recombination ($\Delta E_3$). The time-dependent mechanical properties of DTC8, Six-IC and their blend films were performed by conducting nanoindentation tests (Supplementary Fig. 19 and Supplementary Table 3). As shown in Fig. 4i, the indentation depth increased from D18:DTC8 to D18:Six-IC films under the same applied force, and indicating the branched Six-IC and its BHJ film bears higher ductility[64].

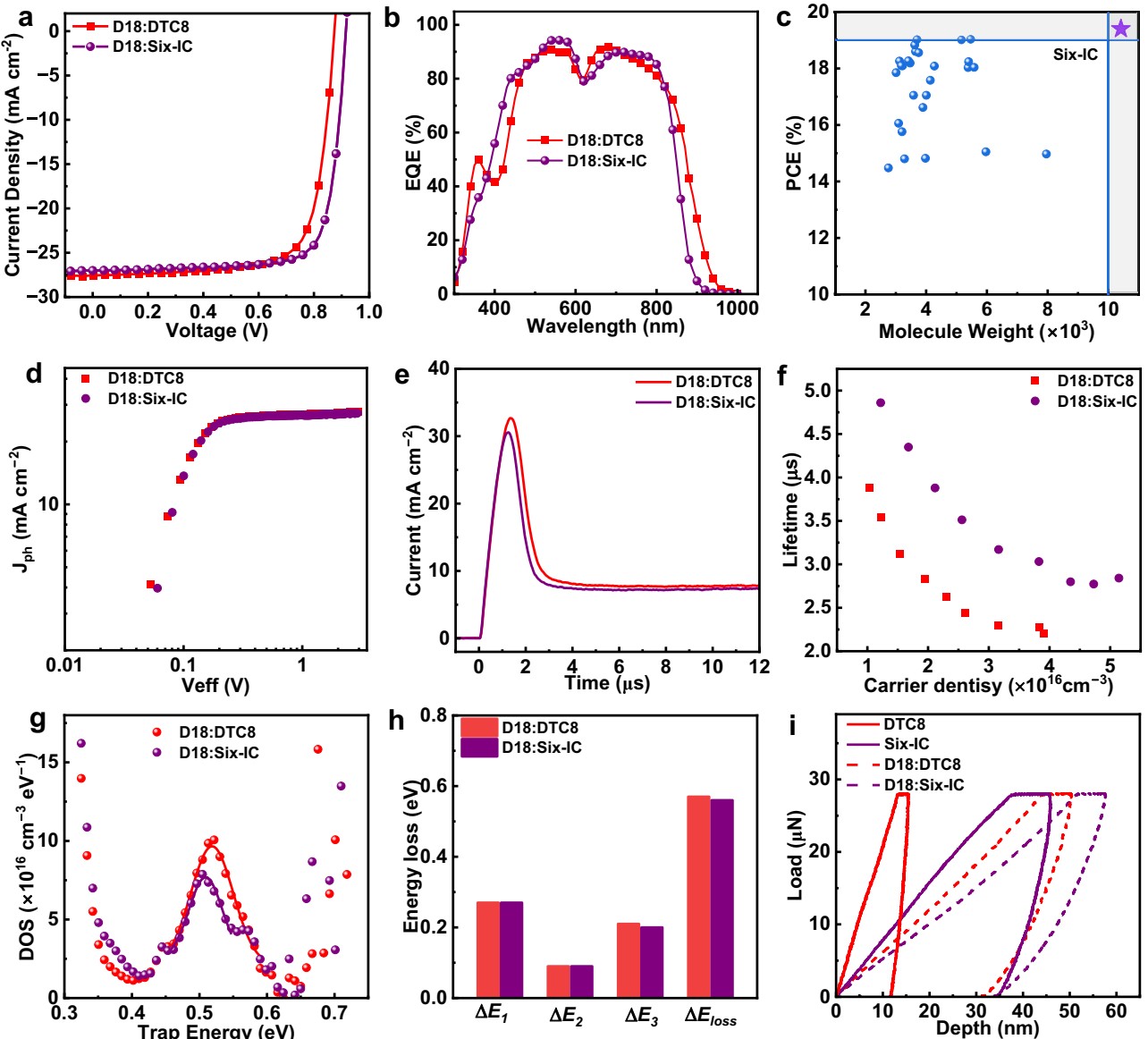

**Fig. 4 | Photovoltaic performance, charge generation, transport, combination loss analysis, and film ductility. a** *J-V* characteristics. **b** EQE spectra. **c** Performance summary. **d** $J_{ph}$-$V_{eff}$ curves. **e** Photo-CELIV measurement. **f** Carrier lifetimes verus carrier densities curves. **g** the trap density measurements by Mott–Schottky method. **h** Detailed energy loss items. **i** The indentation depth verus applied force curves.

**Table 2 | The photovoltaic parameters of D18:DTC8, and D18:Six-IC-based OSCs**

| Donor | $V_{OC}$ [V] | $J_{SC}$ [mA cm⁻²] | FF [%] | PCE ᵃ [%] | $J_{SC}^{EQE}$ [mA cm⁻²] |
|---|---|---|---|---|---|
| D18:DTC8 | 0.86 | 27.5 | 73.9 | 17.6 (17.5 ± 0.1) | 26.8 |
| D18:Six-IC | 0.92 | 27.0 | 78.5 | 19.4 (19.3 ± 0.1) | 26.0 |

ᵃThe average parameters were calculated from 8 independent devices.

Following the discussion, the active layer morphology is investigated through atomic force microscopy (AFM), transmission electron microscopy (TEM), grazing incidence wide-angle X-ray scattering (GIWAXS) experiments by order[65–70]. From Fig. 5a d, we can observe that freshly cast D18:DTC8 and D18:Six-IC blend films display no significant difference, with charge transport favorable nanofibrillar structures. Furthermore, the thermodynamic separating behavior is evaluated by the contact angle experiment on three neat films with water and diiodomethane (DIM) (Supplementary Fig. 20

and Supplementary Table 4). The images with marked contact angle values allow for the calculation of surface free energy for each material and derive a lower χ value of 0.25 K for the D18:Six-IC pair compared to 0.46 K for the D18:DTC8 pair. This trend suggests that Six-IC may induce slightly smaller phase segregation when paired with D18, thereby enhancing the donor-acceptor interfaces. The 2D GIWAXS patterns and corresponding line-cut profiles of in-plane (IP) and out-of-plane (OOP) directions are presented in Fig. 5e h, i. The d-spacing values for IP directional (100) peak and OOP directional

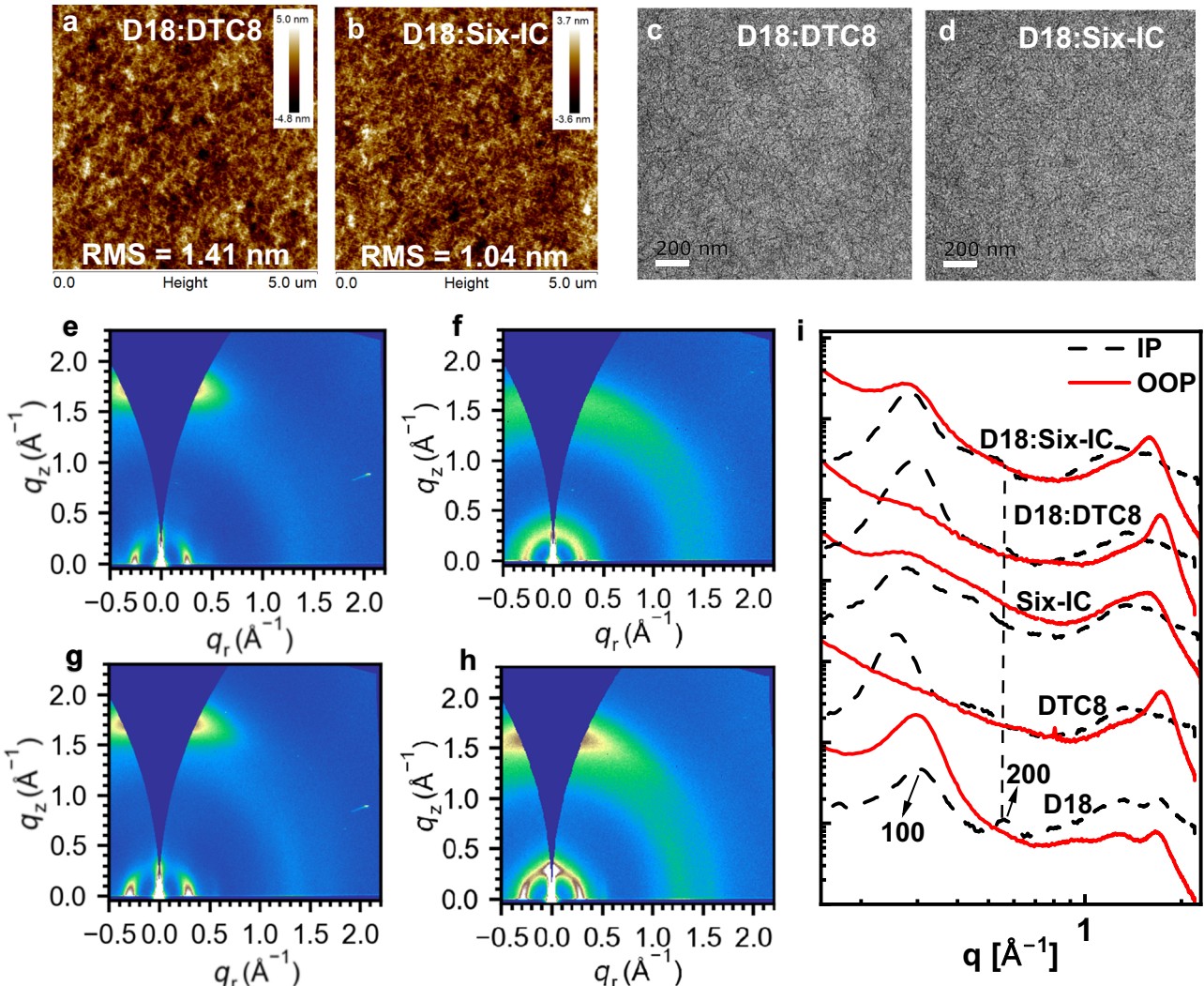

**Fig. 5 | Morphology characterizations. a**, **b** AFM height images of optimal BHJ films. **c**, **d** TEM images of optimal BHJ films. **e**–**h** 2D GIWAXS images. **i** The related line-cut curves.

(010) peak of neat DTC8 film are 24.313 Å and 3.649 Å, related to coherence length (CL) values of 95.960 Å and 26.787 Å. In comparison, these values for neat Six-IC film are 23.173 Å, 4.013 Å, 62.110 Å, and 12.181 Å, respectively (Supplementary Tables 5–6). Interestingly, Six-IC displays a relatively dispersed 010 diffraction peak, primarily indicating multidirectional crystallization in Six-IC, which could potentially lead to multi-dimensional charge transport. In the blend films, D18's strong crystallinity dominates the in-plane (IP) directional signals for both systems (Supplementary Tables 7–8 and Supplementary Fig. 21). The D18:Six-IC blend showed slightly reduced 200 peak intensity in the in-plane (IP) direction compared to D18:DTC8, primarily due to the enhanced miscibility. The slightly lower coherence length (CL) value for the π–π stacking in D18:Six-IC blend suggests a controlled suppression of crystallization.

To further investigate the charge behavior of the two acceptors and their blend films, femtosecond transient absorption spectroscopy (fs-TAS) experiments were conducted[71–74]. Excitation by a 790 nm wavelength laser enabled visible range detection, with 2D contour maps and selective time spectra shown in Fig. 6a h. Based on the signal variations, the features observed in the range of 585–590 nm for Six-IC and 600–610 nm for DTC8 correspond to the crossing points where no distinct acceptor signatures are present, providing an ideal range for detecting hole polaron kinetics in blends (Supplementary Fig. 22). Consequently, the decay kinetics in these regions represent charge

recombination. In addition to polaron features, we also investigated the exciton kinetics in the range of 700–710 nm for Six-IC and 670–680 nm for DTC8. The ultrafast exciton dissociation, coupled with polaron generation, indicates efficient hole transfer in both systems. As depicted by Fig. 6i, D18:Six-IC contains a faster hole transfer, thus leading to quicker exciton dissociation. On the other hand, the range of 600 nm to 610 nm refers to nearly no TAS signal generates from acceptors, so blend film's signals can be utilized to describe the polaron kinetics. Displayed in Fig. 6i as well, the polaron decay after 100 picosecond (ps) clearly indicates a longer charge lifetime, and less recombination happened in D18:Six-IC.

One step further, thermally degraded films depicted in Fig. 7a d demonstrate the suppression of molecular diffusion achieved through the construction of multimer acceptors. The D18:DTC8 binary film is observed with oversized aggregates after annealing at 150 °C for 3 h in photoinduced force microscopy (PiFM) images probed by their characteristic peaks (Supplementary Fig. 23), whereas Six-IC-based film largely retains its initial phase separation characteristics. This assumption is further supported by the AFM height and TEM images (Fig. 7e–h). When annealed at 150 °C for 3 h, the D18:Six-IC films exhibited minimal changes, whereas the D18:DTC8 films showed a significant increase in RMS values from 1.41 nm (Fig. 5a) to 9.17 nm (Fig. 7e) in the AFM images, along with large-scale phase separation in the TEM images (Figs. 5c and 7f). Meanwhile, the OSC device based on

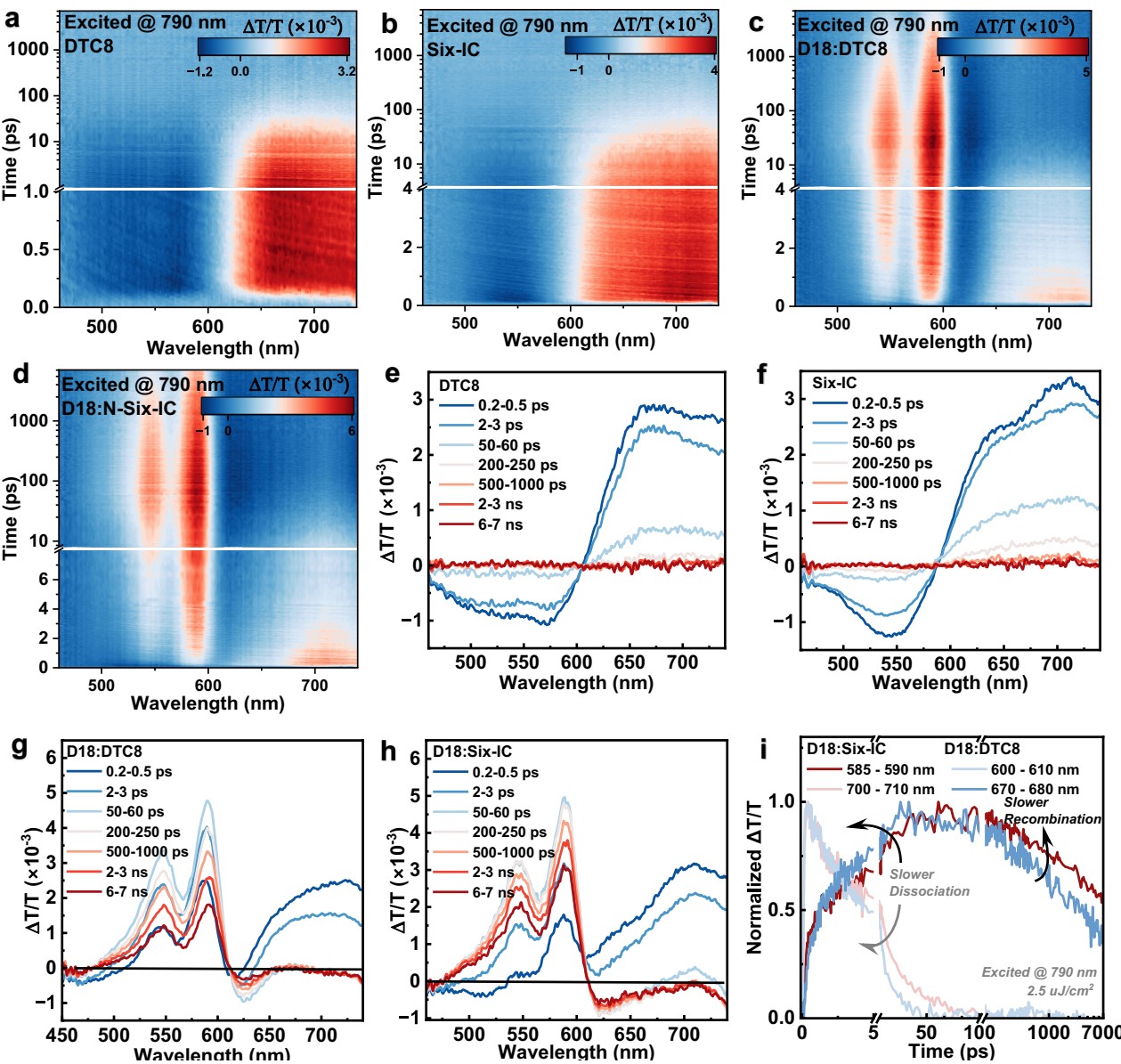

**Fig. 6 | Transient absorption spectroscopy. a–d** 2D contour maps of TAS results. **e–h** Corresponding representative time spectra. **i** Charge kinetics for generation and recombination processes.

D18:Six-IC retained 93.8% of its initial efficiency, whereas the one based on D18:DTC8 retained only 76.2% (Supplementary Fig. 24). The desirable outcome is reasonably attributed to the multiple arms strategy, which induces a higher $T_g$ and enhances intermolecular interactions, thereby stabilizing the morphology. To further assess the stability of real devices, the storage and thermal stability of the two devices were compared, as shown in Fig. 7i. After 1200 h of storage, the D18:Six-IC-based device retains up to 98.0% of its initial efficiency, while the D18:DTC8-based device exhibits a more significant decline, decreasing to 88.2% of its initial efficiency. An even greater distinction is observed in their thermally annealing devices (Fig. 7j). The Six-IC-based device retains up to 94.2% of its initial efficiency, while the DTC8-based device drops to a low value of 67.2% after annealing at 85 °C for 1200 h. These results are supported by the significantly improved $T_g$ of Six-IC and the enhanced homo- and heteromolecular interactions, effectively contributing to the goal of promoting morphology stability by synthesizing acceptor materials with high molecular weight.

In summary, we present a report of a high-performing dendritic hexamer acceptor achieving 19.4% efficiency in binary OSCs through a branched molecule design strategy. The dendritic acceptor, Six-IC, meets the practical requirements of high molecular weight, high synthesis yield, and precise molecular structure. More interestingly, the dendritic acceptor Six-IC exhibits significantly suppressed crystallization, enhanced film ductility, and probable multichannel charge transport compared to its monomer (DTC8). Moreover, due to its increased molecular weight, which enhances $T_g$ and restricts molecular diffusion in blends, the Six-IC-based device demonstrated significantly improved stability compared to the DTC8-based control device. We believe our contribution will provide valuable guidance for future acceptor design toward commercialization.

## Methods

### Materials

3,9-diundecyl-12,13-dihydro-[1,2,5]thiadiazolo[3,4-e]thieno[2″,3″:4′,5′]thieno[2′,3′:4,5]pyrrolo[3,2-g]thieno[2′,3′:4,5]thieno[3,2-b]indole (compound 1), PNDI-F₃N, and D18 were purchased from Solarmer Energy, Inc. 1,8-Dibromooctane, 11-(bromomethyl)tricosane, 10,15-dihydro-5H-tribenzo[a,d,g][9]annulene-2,3,7,8,12,13-hexaol,    and

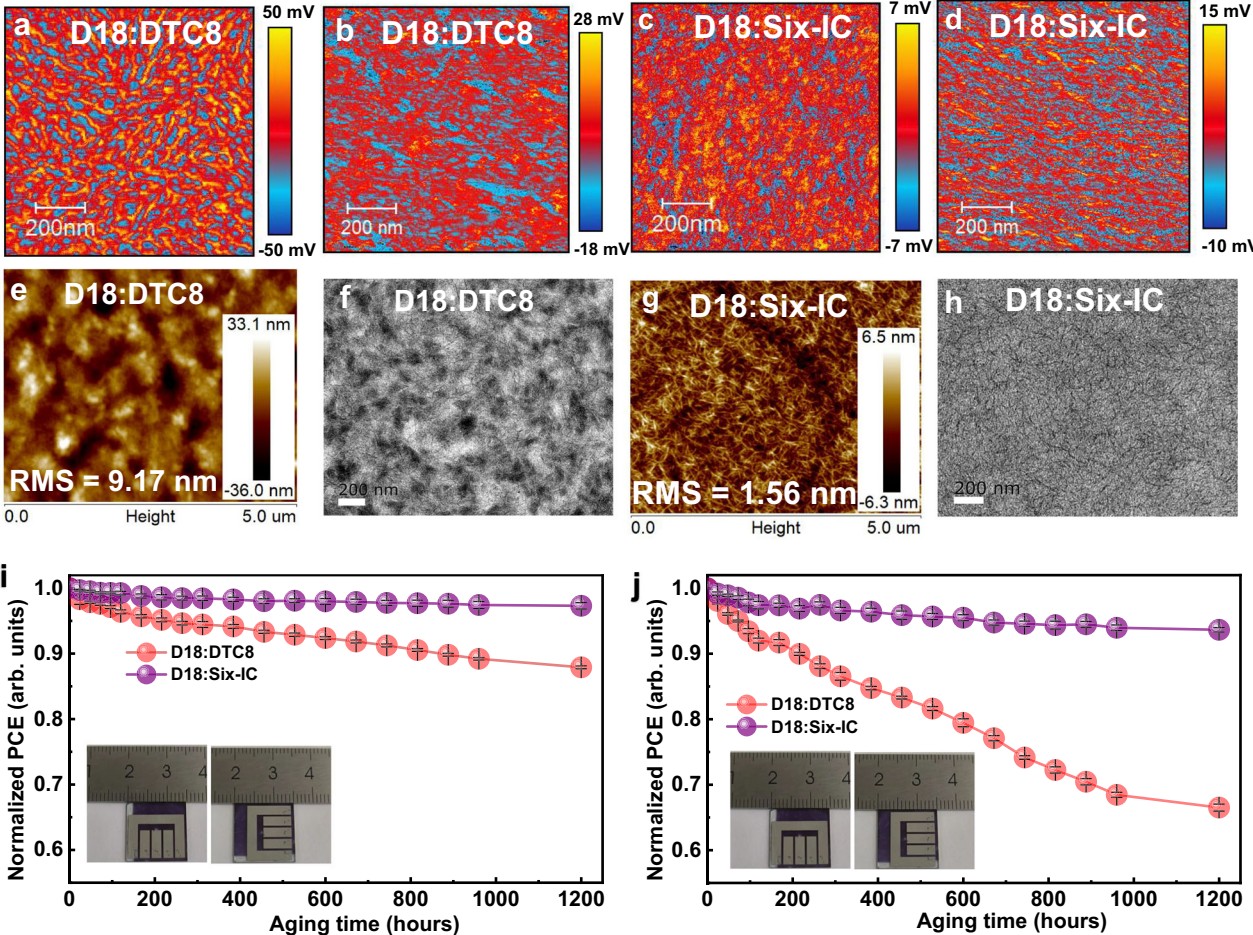

**Fig. 7 | Stability characterizations.** Photoinduced force microscopy (PiFM) images: (**a**, **c**) without annealing and (**b**, **d**) after annealing at 150 °C for 3 h. AFM (**e**, **g**) and TEM (**f**, **h**) images of the related blend film after annealing at 150 °C for 3 h. The degradation of the related devices during (**i**) storage and (**j**) heating at 85 °C, obtained from the average of four devices with the error depicted as standard deviation.

2-(5,6-difluoro-3-oxo-2,3-dihydro-1H-inden-1-ylidene)malononitrile were purchased from Bide Pharmatech. Ltd. Other chemicals and solvents were purchased from commercial sources (Sigma Aldrich, Acros, Strem, or Alfa Aesar) and used as received.

### Device fabrication and test

The conventional ITO/PEDOT/Active layer/PNDIT-F3N/Ag structure was used to fabricate the OSCs. Indium tin oxide (ITO) substrates were cleaned sequentially by sonication in detergent, deionized water (DIW), and isopropanol. After drying in an oven at 60 °C overnight, the substrates were treated with oxygen plasma for 5 min, followed by spin-coating with PEDOT(CLEVIOS P VP Al 4083) at 3000 rpm for 30 s. For the active layer, chloroform (CF) was used as the processing solvent, with the blend solution concentration fixed at 8.05 mg/mL (donor:acceptor = 1:1.3 (in wt)). The thickness was controlled at ~100 nm by adjusting the spin speed. The additive was removed by vacuum drying for 1 h. For thermal stability studies and the related morphological tests, the active layer films were further annealed at 150 °C for 3 h. A 5 nm layer of PNDIT-F3N (0.5 mg/mL in methanol) was spin-coated onto the active layer as the cathode interface. Finally, a 100 nm layer of silver was thermally deposited on top of the interface through a shadow mask in a vacuum chamber at a pressure of $1 \times 10^{-7}$ mbar. The device's effective area was confined to 0.04 cm² using a non-reflective mask to improve measurement accuracy. The device contact area was 0.057 cm², and the device illuminated area during testing was 0.033 cm², determined by a mask. The current density-voltage (*J-V*) characteristics were measured using a computer-controlled Keithley 2400 source meter under 1 sun illumination from an AM 1.5 G solar simulator (Enlitech SS-F5, Taiwan). The light intensity was calibrated using a standard silicon solar cell (certified by the China General Certification Center) to 100 mW/cm² before testing the *J-V* characteristics. The external quantum efficiency (EQE) spectra were recorded using a QE-R measurement system (Enlitech, QE-R3011, Taiwan).

### General characterizations

UV-Vis absorption spectra were recorded using a SHIMADZU UV-3600 spectrophotometer with corrections for quartz absorption. Ultraviolet photoelectron spectroscopy (UPS) measurements were recorded using a RIKEN KEIKI spectrometer (Model AC-3). Inverse photoemission spectroscopy (IPES) measurement was performed using a customized ULVAC-PHI LEIPS instrument with Bremsstrahlung isochromatic mode. Photo-CELIV measurements were performed using the Keysight E4990A Vector Network Analyzer. The experiment involved determining the extracted charge carrier density as a function of delay time, with a ramp rate of 200 V/ms. The transient photovoltage (TPV) technique using PAIOS was based on monitoring the photovoltage decay after a small optical perturbation under various constant bias light intensities. A small optical perturbation (<3% of the $V_{OC}$, ensuring $\Delta V_{OC} \ll V_{OC}$) was applied, and the resulting voltage decay was recorded

to directly monitor nongeminate charge carrier recombination. The photovoltage decay kinetics for all devices followed a mono-exponential decay: $\delta V = A\exp(-t/\tau)$, where $t$ is the time, and $\tau$ is the charge carrier lifetime. Charge extraction (CE) measurements were conducted using a Keysight E4990A Vector Network Analyzer. Devices were illuminated at varying light intensities and kept at open-circuit conditions. After turning off the light, the voltage was set to zero or taken to short-circuit conditions within a few hundred nanoseconds to extract charges, and the current was integrated to determine the number of extracted charges. Capacitance-frequency measurements were conducted using a commercially available PAIOS system. These measurements were performed in the dark under an applied bias corresponding to the open-circuit voltage of the OPV, with a frequency range from 10 MHz to 1 Hz. Contact angle measurements were performed using a water or diiodomethane contact angle measurement system (OCA40 Micro). Atomic force microscopy (AFM) of the blended films was performed using a Digital Instruments DI Multimode Nanoscope III in tapping mode. The samples for the AFM measurements were prepared as the same conditions for OSC devices. Transmission electron microscopy (TEM) characterization was conducted with a JEM-2100F instrument. 2D-GIWAXS measurements were carried out on an XEUSS 3.0 UHR SAXS/WAXS system (XENOCS, France) with a Eiger2 R 1 M 2D detector featuring 0.075 × 0.075 mm active pixels in integration mode. The detector was positioned 100/2000 mm downstream from the sample, and the precise sample-to-detector distance was calibrated using a silver behenate standard. The Cu incident X-ray (8 KeV) with a 0.9 × 0.9 mm or 0.5 × 0.5 mm spot provided sufficient q-space.

## Reporting summary

Further information on research design is available in the Nature Portfolio Reporting Summary linked to this article.

## Data availability

The data that support the findings of this study are presented in Supplementary Information, Source Data, and Supplementary Data files. The source data for Fig. 4a, Table 1, Supplementary Fig. 24a, b generated in this study are provided as a Source Data file. The source data for Fig. 3a–d are provided as Supplementary Data file. Source Data and Supplementary Data are provided with this paper. Source data are provided with this paper.

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

## Acknowledgements

T.J. acknowledges the financial support from Natural Science Foundation of China (No. 21805099). S.L.(Shengjian Liu) acknowledges the financial support from Natural Science Foundation of China (No. 21805097), the Guangdong Natural Science Foundation (No. 2021B1515120073), and the Guangdong Provincial Science and Technology Foundation (No. 2022A0505050068). J.W. thanks the Guangdong government and the Guangzhou government for funding (2021QN02C110), the Guangzhou Municipal Science and Technology Project (No. 2023A03J0097 and No. 2023A03J0003), and National Natural Science Foundation of China (52303249). J. W. also thanks the support of HKUST Materials Characterization and Preparation Facility Guangzhou (MCPF-GZ). R.M. gratefully acknowledges the support from PolyU Distinguished Postdoctoral Fellowship (1-YW4C). The authors acknowledge the Green e Materials Laboratory and the HKUST Materials Characterization and Preparation Facility (MCPF) Guangzhou (GZ) for their facilities and technical support.

## Author contributions

T.J., T.L., and Y.Y. contributed equally to this work. T.J., R.M., J.W., and S.L.(Shengjian Liu) proposed the research, designed the experiments, and supervised the project. T.J. synthesized the Six-IC acceptor. T.L., Y.Y., L.W., and R.M. fabricated the BHJ OSCs. Y.H. conducted the simulated calculation. Y.L. (Yongmin Luo) carried out the experiments of GIWAXS measurements and analyzed the TAS data. H.C., Z.Z., K.L., T.A.D.P., and Y.L. (Yao Li) helped to analyze the data. S.L. (Sha Liu), J.Z., C.L., J.C., and F.H. provided the experiment conditions. T.J. and R.M. wrote the manuscript. All authors commented on the manuscript.

## Competing interests

The authors declare no competing interests.
