## [Peer review file · Nature Communications]

A Dendritic Hexamer Acceptor Enables 19.4% Efficiency with Exceptional Stability in Organic Solar Cells

Corresponding Author: Dr Ruijie Ma

Version 0:

Reviewer comments:

Reviewer #1

(Remarks to the Author)

This manuscript presents the design and synthesis of a novel multi-armed macromolecular hexamer acceptor using a branch-connecting strategy. The acceptor is produced with a high yield of over 58%, featuring an ultra-high molecular weight (MW > 10000), which is generally considered the threshold between small molecules and macromolecules. The macromolecular acceptor exhibits modulated crystallinity, enhanced miscibility, and improved charge transport via additional channels between the armed units. Consequently, an impressive efficiency of 19.4% was achieved in organic solar cells, placing it among the top-performing OSCs. Furthermore, the high molecular weight of Six-IC significantly improves the glass transition temperature (T_g). It strengthens both homo- and hetero-molecular interactions, leading to outstanding device stability and film ductility. In addition, the molecular design strategy is novel and unique, with reasonably high synthetic challenge. Overall, it is solid and constructive work for the OSC community, and I believe this paper should be accepted after addressing the following issues:

1. Given the unique structure of the Six-IC molecule reported in the article, I suggest that the authors use the term "dendritic molecule" instead of "macromolecule" or "giant molecule," as it more accurately reflects the characteristics of this molecule.
2. I noticed that in the mass spectrum, there is a relatively strong fragment peak on the left side of the molecular ion peak of Six-IC. Could this indicate the presence of an impurity? If not, could you clarify the origin of this peak? In addition, I recommend that the authors show the purity of the Six-IC using the HPLC system.
3. From a chemical perspective, the difficulty of purification significantly impacts the feasibility of large-scale production and commercialization. The macromolecular acceptor reported in this article has a significant molecular weight, with the intermediate Six-H featuring six arms and the final product having 12 terminal groups. How were these compounds purified? Were there any specific challenges encountered during the purification process? It could be even better if the authors further clarified the overall yield of 58% of the Six-IC is highly impressive through citations or a list of supplementary tables.
4. Regarding the DSC results, the traces for Six-IC and DTC8 only display the heating process. What about the cooling process? Are there any noticeable crystallization peaks observed during cooling?
5. As far as I know, polymerized small molecule acceptors (PSMAs) reported in the literature can also produce highly stable and efficient solar cell devices. What are the advantages of the materials designed in this article? The main text should include a more detailed discussion on this topic.
6. Interestingly, the solution of Six-IC exhibited an additional absorption band at 689 nm, attributed to the "distinctive packing of the SMA arms." However, could the authors confirm whether this is due to the formation of H-aggregates of the DTC8 moiety in solution? Additionally, the authors should explain the reasons behind the significant blue shift observed for Six-IC. Based on my understanding, the use of dendrimer to prevent molecular aggregation has been well-reported in dendrimer iridium(III), platinum(II) and gold (III) OLED (<https://doi.org/10.1002/anie.201206457>; <https://doi.org/10.1021/ja903157e>). So, I also suggest that the authors delete the phrase "could be a new approach to regulate molecular aggregation and morphology characteristics" instead of citing the corresponding references in related research fields.

7. As stated by the authors in the article, "This indicates that Six-IC and DTC8 follow different aggregation motifs in liquid and solid states, supporting our hypothesis that the designed molecular configuration for GMAs could be a new approach to regulate molecular aggregation and morphology characteristics." Could the authors provide temperature-dependent absorption spectra of Six-IC and DTC8 in dilute solution to further explore their distinct aggregation behaviors?
8. The data in Supplementary Table 1 need to be consistent with the main text. The authors should double-check and correct this discrepancy.
9. According to the GIWAXS patterns, Six-IC exhibits a less oriented molecular packing motif, which is generally considered unfavorable for charge transport. How do the authors address this issue?
10. The PiFM results are intriguing and informative. However, additional measurement details should be provided, such as the characteristic FT-IR peaks of D18, DTC8, and Six-IC.
11. Some sentences, including but not limited to "which is unprecedented", should be deleted to avoid exaggeration.
12. On page 8, during the author's discussion of the DFT calculation, they mentioned that the two SAM units are considered for use in the calculation. The reason for that is time-saving. I suggest the author note that the electronic structure is not altered by SMA units, and the simulation is not affected by reducing the molecular complexity, which should be the better reason for this compromise between calculation time and simulation accuracy. Please edit accordingly.

Reviewer #2

(Remarks to the Author)

The manuscript reports a new macromolecular hexamer acceptor with a branch-connecting strategy, resulting in a molecular weight exceeding 10 kDa. The binary OSCs based on D18 achieve an efficiency of 19.4%. While the study presents a new case for the use of high-molecular-weight acceptors in OSCs, a more detailed mechanistic understanding of the structure-property relationship and the advantages of this design, would enhance its quality. Also, several critical issues should be addressed for further investigation:

- (1) The authors state in the MD simulations, "To save time, only two SMA subunits on Six-IC were used for the calculations." However, for a high-molecular-weight acceptor with the cup-shaped conformation of cyclotrimeratrylene, the number of SMA subunits is crucial for understanding the intramolecular packing behavior. As a result, the current DFT calculations lack scientific significance and cannot serve as a reliable reference.
- (2) The significant blue-shift observed in the Six-IC films, approximately 46 nm relative to DTC8, typically suggests an elevation in the ELUMO energy level. How, then, can the authors explain the nearly identical ELUMO levels of Six-IC (-3.92 eV) and DTC8 (-3.91 eV)? Also the blue shift in the absorption should be explained for the new acceptor.
- (3) It is widely accepted that the Voc values of OSC devices are generally positively correlated with the difference between the EHOMO of the donor and the ELUMO of the acceptor. However, the significant difference in Voc values for Six-IC and DTC8-based devices, despite their similar ELUMO values, is not sufficiently explained by the energy loss analysis presented in Figure 3h.
- (4) In the fs-TAS experiments, the authors state that "D18 contains a faster hole transfer, thus leading to quicker exciton dissociation toward boosted Jsc" and that "a longer charge lifetime and less recombination occur in D18." However, the observed lower Jsc for D18 devices compared to D18 challenges this conclusion and necessitates a more compelling explanation.
- (5) During device fabrication, the blend films typically exist in a thermodynamically metastable state to achieve higher efficiency, resulting in a stability curve that usually exhibits a "burn-in" loss stage. Interestingly, both Six-IC and DTC8-based devices show no such "burn-in" loss stages. This represents a significant advancement in organic photovoltaics by eliminating the "burn-in" loss, which should be highlighted and well explained in the abstract. If the case, this can be compared with other results? Additionally, the manuscript should provide detailed information regarding the stability testing methodology, including the model of the stability characterization equipment used.
- (6) The figures should standardize line thickness, font size, and formatting to ensure consistency and clarity.
- (7) The significant progress in OPV, particularly with ITIC and its derivatives, should be supported by relevant references that highlight not only advancements in efficiency but also important milestone works, as well as the mechanistic studies.
- (8) The synthesis of supermolecules remains challenging, yet the yields (over 80-90%) achieved in this work are significantly higher than those in previous reports, even with a molecular weight exceeding 10 kDa. This should be thoroughly explained for readers, supported with detailed comparisons to prior studies, and highlighted in the manuscript.
- (9) For the DSC curves of Six-IC, to check the result of TC values, different scan rates should be proved. The melting enthalpy (ΔH_m) of Six-IC significantly decreased from 23.85 J/g to 9.07 J/g. How can this change be explained effectively? This should be proved the manuscript. The authors states "This result was further confirmed by DSC. How can this be explained?"
- (10) The authors state "the enhanced miscibility resulting from increased donor/acceptor interface contact" from GIWAXS, how can this be explained?
- (11) In Figure 6, the evidence for morphology stability is provided. However, is there any significant difference between the morphologies of the two blends based on this data? This should be discussed with the PCEs.
- (12) D18:Six-IC blend films shows high mobility while its crystallization is greatly suppression. How can this be explained?
- (13) The 13C of all the new materials should be provided. also the TGA.

Reviewer #3

(Remarks to the Author)

The authors present a macromolecular hexamer acceptor with a molecular weight of 10k g/mol. This hexamer addresses the molecular weight limitations of GMAs and achieves a high production yield exceeding 58%. The macromolecular acceptor, Six-IC, demonstrates enhanced crystallinity and miscibility with the donor, leading to superior morphology compared to its monomer counterpart, DTC8. The binary OSCs based on the D18 system exhibit a high efficiency of 19.4% for high-molecular-weight acceptors, alongside notable device stability and film ductility.

Extensive characterization of the molecular structure and PV devices was performed to support the conclusions, supplemented by DFT and MD simulations, which are presented clearly for readers' comprehension. However, the reviewer remains concerned that numerous studies on similar GMA concepts have been published in the past three years. Despite this concern, the manuscript's structure and related characterization data are well-organized. As the authors noted, the high molecular weight and production yield achieved in this work have the potential to advance the GMA field, alongside a 19.4% efficiency and improved thermal stability. Enclosed are some suggestions for improving the manuscript before publication:

- 1). The authors list Molecular Weight vs. PCE in Figure 3c to highlight the novelty of this work. As the authors also claim that the synthesis of Six-IC is at a high yield level among GMAs, the reviewers suggest plotting a figure on the related synthesis yield. Additionally, the authors could provide some discussions on why the yield is high for such high molecular weight acceptor. This would help readers understand the rationale behind this GMA, which is related to the novelty of this work.
- 2). Figure 1g shows the DSC curves of Six-IC; however, the $T_c = 178^\circ\text{C}$ is not convincing, as the DSC curve rises after 150°C . Rechecking and remeasuring the DSC would help clarify this issue.
- 3). As the authors measured the energy levels by CV, it would be preferable to confirm the HOMO/LUMO by UPS/IPES, as CV measurements can be challenging in determining the HOMO/LUMO of these PV materials.
- 4). Figures 6i and 6j present the thermal stability of the devices, seemingly measured PV devices after different certain aging periods. It would be helpful to include the related sample size and error bars, along with averaged or maximum values, for clearer representation.

Some minor comments:

- 5). The unit of molecular weight is missing in the manuscript. "g/mol" should be added before 10,000, or alternatively use "kDa".
- 6). In Table 1, since $M_n/M_w/PDI$ only applies to polymers, for an organic compound with a defined molecular weight ($M_n = M_w, PDI = 1$), using M (g/mol) would be more appropriate.
- 7). There are some overly promotional statements, e.g.: i) Page 4, the last sentence of the Introduction: "ultra-high molecular weight." ii) Page 17, the last sentence: "advance the entire OSC field toward commercialization." This work focuses on NFAs, and "advance the entire OSC field toward commercialization" seems overstated.
- 8). Some typos: in the caption of Fig. 2: "(c) Charge Density Differences" (all capitalized). On page 17, Materials section: "PNDI-F3N," where the number 3 is subscripted.
- 9). Open question: Did the authors consider using fully rigid aromatic structures as the core, or was this approach abandoned due to low synthesis yield? It appears that three flexible -CH₂- linkers were used in Six-IC. A fully rigid aromatic core with six arms would likely provide greater thermal stability, which is typically preferred for such structures in the OPV field.

Reviewer #4

(Remarks to the Author)

Version 1:

Reviewer comments:

Reviewer #1

(Remarks to the Author)

The author has carefully supplemented the experiments to address my concerns and comments, demonstrating a thorough understanding of the subject. In addition to making the necessary adjustments, the author has provided detailed explanations and justifications for the changes. This enhances the robustness of the research and adds clarity to the overall findings. With these revisions, the current version is well-prepared and meets the required standards for acceptance by Nature Communications.

Reviewer #2

(Remarks to the Author)

After revision, the manuscript can be accepted for publication.

Reviewer #3

(Remarks to the Author)

Glad to see that authors took their time to address all reviewers comments and improved their manuscript. Several data are added and it looks way more improved and can be accepted.

Reviewer #4

(Remarks to the Author)

29-November-2024

Via this letter, we submit our revisions and responses to the suggestions made by the referees, with regard to our manuscript entitled “A Macromolecular Hexamer Acceptor Enables a Highly Efficient Organic Solar Cell with 19.4% Efficiency and Excellent Stability” (No. NCOMMS-24-59659) for publication as a research article in Nature Communications. In a correspondence received on Oct. 22th, 2024, our manuscript was deemed suitable for publication in Nature Communications, and we were invited to resubmit a revised manuscript file in light of the suggestions of the reviewers. We are grateful to the referees for their constructive feedback and the time spent reviewing our work. We have carefully considered the comments and suggestions of the reviewers, and you may find below the details of our response:

Response to the reviewers' comments

REVIEWER COMMENTS

Reviewer #1 (Remarks to the Author):

This manuscript presents the design and synthesis of a novel multi-armed macromolecular hexamer acceptor using a branch-connecting strategy. The acceptor is produced with a high yield of over 58%, featuring an ultra-high molecular weight ($MW > 10000$), which is generally considered the threshold between small molecules and macromolecules. The macromolecular acceptor exhibits modulated crystallinity, enhanced miscibility, and improved charge transport via additional channels between the armed units. Consequently, an impressive efficiency of 19.4% was achieved in organic solar cells, placing it among the top-performing OSCs. Furthermore, the high molecular weight of Six-IC significantly improves the glass transition temperature (T_g). It strengthens both homo- and hetero-molecular interactions, leading to outstanding device stability and film ductility. In addition, the molecular design strategy is novel and unique, with reasonably high synthetic challenge. Overall, it is solid and constructive work for the OSC community, and I believe this paper should be accepted after addressing the following issues:

Response: We thank the reviewer for the positive comments and recommendation for publication of this work.

1. Given the unique structure of the Six-IC molecule reported in the article, I suggest that the authors use the term "dendritic molecule" instead of "macromolecule" or "giant molecule," as it more accurately reflects the characteristics of this molecule.

Response: We thank the reviewer for the insightful comments and suggestions. We fully agree that the term "dendritic molecule" more accurately reflects the characteristics of the Six-IC molecule compared to "macromolecule" or "giant molecule." Following this recommendation, we have replaced "macromolecule" and "giant molecule" with "dendritic molecule" in the descriptions of Six-IC and have highlighted these changes accordingly.

2. I noticed that in the mass spectrum, there is a relatively strong fragment peak on the left side of the molecular ion peak of Six-IC. Could this indicate the presence of an impurity? If not, could you clarify the origin of this peak? In addition, I recommend that the authors show the purity of the Six-IC using the HPLC system.

Response: We thank the reviewer for the valuable comments and suggestions. The mass spectrum was remeasured under optimized conditions and updated in the main text. The fragment peaks observed to the left of the molecular ion peak of Six-IC are primarily attributed to the loss of the IC terminal group. This occurs due to the relatively weak double bond connecting the terminal group in the Six-IC molecule. Similar fragmentation behavior has also been reported in several studies in the literature (*Adv. Funct. Mater.* **34**, 2400810 (2024); *Nat. Energy* **7**, 1180–1190 (2022); *Angew. Chem. Int. Ed.* **62**, e202308595 (2023)). we Moreover, to further confirm the structure, the ¹³C NMR spectra of both the intermediates and Six-IC (**Supplementary Fig. 16, Fig. 19, and Fig. 22**), along with the high-temperature GPC analysis of Six-IC (**Supplementary Fig. 24**), have been performed and are now included in the revised **Supplementary Information**.

Supplementary Fig. 16. ^{13}C NMR spectrum of **Six-H** in CDCl_3 .

Supplementary Information, Page 21

Supplementary Fig. 19. ^{13}C NMR spectrum of **Six-CHO** in CDCl_3 .

Supplementary Information, Page 22

Supplementary Fig. 22. ^{13}C NMR spectrum of **Six-IC** in $\text{C}_2\text{D}_2\text{Cl}_4$ at $100\text{ }^\circ\text{C}$.

Supplementary Information, Page 23

18. Gel Permeation Chromatography Measurement

The molecular weights of **Six-IC** was obtained on an Acquity Advanced Polymer Chromatography (Waters) with a high-temperature chromatograph in 1,2,4-trichlorobenzene at $150\text{ }^\circ\text{C}$ and using a calibration curve of polystyrene standards.

Workbook Details

Eluent: TCB stabilised with 0.0125% BHT
Column Set: PLgel MIXED-B LS 300x7.5mm x2
Detector: RI
Flow Rate: 1.00 ml/min
Temperature: 150
Injection Volume: 200.0 ul

Analysis Using Method: 20240901

Comments:

Calibration Used: 2024/9/7 10:39:10

High Limit MW RT: 10.95 mins
High Limit MW: 6217712
K: 17.5000

Low Limit MW RT: 17.27 mins
Low Limit MW: 623
Alpha: 0.6700

**MW Averages**

Peak No	Mp	Mn	Mw	Mz	Mz+1	Mv	PD
1	11942	10001	10974	11844	12624	10792	1.09729

Processed Peaks

Peak No	Name	Start RT (mins)	Max RT (mins)	End RT (mins)	Pk Height (mV)	% Height	Area (mV.secs)	% Area
1		15.18	15.60	16.50	61.8949	100	1700.27	100

Supplementary Fig. 24. High-temperature GPC curve of Six-IC.

3. From a chemical perspective, the difficulty of purification significantly impacts the feasibility of large-scale production and commercialization. The macromolecular acceptor reported in this article has a significant molecular weight, with the intermediate Six-H featuring six arms and the final product having 12 terminal groups. How were these compounds purified? Were there any specific challenges encountered during the purification process? It could be even better if the authors further clarified the overall yield of 58% of the Six-IC is highly impressive through citations or a list of supplementary tables.

Response: We thank the reviewer for their insightful comments and suggestions. We completely agree that the difficulty of the purification process significantly impacts

the feasibility of large-scale production and commercialization. Although the synthesis of the intermediate Six-H involves the attachment of six arms, the nucleophilic reaction between brominated alkanes and phenolic hydroxyl groups is straightforward and typically proceeds smoothly, usually achieving a high yield of over 95% (*Tetrahedron Lett.*, **19**, 4581 (1979); *Tetrahedron* **36**, 1223 (1980); *Chemistry Select* **2**, 4892 (2017); *Molecules* **2**, 602 (2020)). This high reaction efficiency contributes to the overall high yield of Six-H. As shown in the thin-layer chromatography (TLC) plate of the reaction solution from compound **2** to Six-H (Fig. R1), Six-H can be obtained with a high yield, and the excess starting material, compound **2**, can also be recycled. There are two key considerations: First, the reaction must be conducted using an ultra-dry solvent; second, CVT is unstable to oxygen in the presence of alkali, so the reaction should be carried out in an inert gas atmosphere. Additionally, the strong polarity of the phenolic hydroxyl groups contributes to the high overall polarity of the cyclotrimeratrylene core. As compound **2** gradually reacts with the phenolic hydroxyl groups, the molecular polarity steadily decreases. This change in polarity facilitates tracking the reaction progress and simplifies the subsequent purification process of Six-H. Although the attachment of 12 aldehydes and IC terminals to Six-CHO and Six-IC, respectively, may seem complex, the conversions from Six-H to Six-CHO and from Six-CHO to Six-IC were efficiently achieved using the Vilsmeier-Haack and Knoevenagel condensation reactions. These well-established, high-yield methods are widely employed in the synthesis of semiconductor materials (*Green Chem.* **10**, 873 (2008); *Green Chem.* **10**, 767 (2008); *Inorg. Chem.* **59**, 15, 10578 (2020); *Dalton Trans.* 50, 4445 (2021); *J. Am. Chem. Soc.* **144**, 8, 3653 (2022)). Moreover, the branching linkage strategy ensures that the chemical reactivity of each arm remains independent, facilitating efficient reactions and contributing to the high yield of the final product. The use of these reactions to design multi-armed acceptors has been well-documented and typically proceeds smoothly, consistently yielding the target product in high yield (~90%) (*Nat. Commun.* **14**, 2926 (2023); *Angew. Chem. Int. Ed.* **63**, e202400590 (2024); *Adv.*

Energy Mater. **14**, 2400938 (2024)). From **Six-H** to **Six-CHO**, the introduction of polyaldehydes significantly increased the molecular polarity, which, in combination with the high yield of the Vilsmeier-Haack reaction, facilitated easy purification of the polyaldehyde products. In the conversion from **Six-CHO** to **Six-IC**, the starting material, **Six-CHO**, is a slightly viscous solid with excellent solubility, even in *n*-hexane. However, upon attaching the end groups, **Six-IC** becomes insoluble in *n*-hexane or a mixed solvent of *n*-hexane and dichloromethane. A simple column chromatography procedure allows for the easy removal of any remaining impurities (**Fig. R1**) by stirring and washing with *n*-hexane, followed by a mixed solvent of *n*-hexane and dichloromethane. We agree with the reviewer that the overall yield of 58% for **Six-IC** is highly impressive and stands as one of the key highlights of this work. Following the suggestion, we have added the overall yields of representative GMAs to **Supplementary Table 1** and the following discussion was added into the revised main text. Moreover, to ensure better reproducibility, we have provided a more detailed synthesis process for the key intermediate **Six-H** in the **Supplementary Information**, and the updates have been highlighted.

Fig. R1. The chemical structures of Com 2, IC, Six-H, Six-CHO and Six-IC, along with the TLC retention factors.

“This value represents a relatively high yield for synthesizing high-molecular-weight GMAs (Supplementary Table 1), underscoring the potential of the branched molecule design strategy for acceptor synthesis and future applications.”

4. Regarding the DSC results, the traces for Six-IC and DTC8 only display the heating process. What about the cooling process? Are there any noticeable crystallization peaks observed during cooling?

Response: We thank the reviewer for the comments and suggestions. In fact, the DSC has been conducted in both heating and cooling processes. As shown in Fig. R2, within the test range of 0~300°C, we did not observe any significant crystallization peaks during the cooling process.

Fig. R2. DSC curves of DTC8 and Six-IC measured at a heating or cooling rate of 10°C/min.

5. As far as I know, polymerized small molecule acceptors (PSMAs) reported in the literature can also produce highly stable and efficient solar cell devices. What are the advantages of the materials designed in this article? The main text should include a more detailed discussion on this topic.

Response: We thank the reviewer for the comments and suggestions. Polymerized small molecule acceptors (PSMAs)-based all-PSCs can also produce highly stable and efficient solar cell devices. However, PSMAs face several significant drawbacks: (1)

efficient PSMA typically require isomeric purification of IC terminals and a lengthy synthetic pathway, adding considerable complexity to their production; (2) the lack of a defined structure results in significant batch-to-batch variability (*Nano Energy* **72**, 104718 (2020); *Joule* **4**, 1070-1086 (2020); *Adv. Energy Mater.* **12**, 2103193 (2022)), posing challenges for commercial applications. In contrast, our newly developed acceptor, Six-IC, offers several key advantages: (1) a well-defined molecular structure with no batch variability, ensuring high reproducibility; (2) an alkyl chain branching strategy that facilitates high-yield synthesis, making it suitable for commercial production; (3) a dendritic molecular structure that supports multi-dimensional charge transport, potentially enhancing device efficiency; and (4) when compared to PSMA, the efficiency of devices fabricated with our dendritic molecule ranks among the highest reported in PSMA-based PSCs. In line with this suggestion, the relevant discussion has been added to the revised main text and highlighted accordingly.

Main text, Page 3

“benefiting from the innovation of polymerized small molecule acceptors (PSMAs)”

Main text, Page 3

“PSMAs face several notable drawbacks: (1) Efficient PSMA typically require isomeric purification of IC terminals and a lengthy synthetic pathway, adding considerable complexity to their target production. (2) Due to the lack of a defined structure, PSMA experience significant batch-to-batch variability,³³⁻³⁵ creating challenges for future commercial production”

Main text, Page 3

“developing acceptor materials with high yields, reproducibility, and substantial molecular weight presents a critical opportunity for advancing OSCs towards future commercialization.”

Main text, Page 23

*33. Jia T, et al. 14.4% efficiency all-polymer solar cell with broad absorption and low energy loss enabled by a novel polymer acceptor. *Nano Energy* **72**, 104718 (2020).*

34. Wang W, et al. *Controlling Molecular Mass of Low-Band-Gap Polymer Acceptors for High-Performance All-Polymer Solar Cells. Joule 4, 1070-1086 (2020).*
35. Jia J, et al. *Fine-Tuning Batch Factors of Polymer Acceptors Enables a Binary All-Polymer Solar Cell with High Efficiency of 16.11%. Adv. Energy Mater. 12, 2103193 (2022).*

6. Interestingly, the solution of Six-IC exhibited an additional absorption band at 689 nm, attributed to the “distinctive packing of the SMA arms.” However, could the authors confirm whether this is due to the formation of H-aggregates of the DTC8 moiety in solution? Additionally, the authors should explain the reasons behind the significant blue shift observed for Six-IC. Based on my understanding, the use of dendrimer to prevent molecular aggregation has been well-reported in dendrimer iridium(III), platinum(II) and gold (III) OLED (<https://doi.org/10.1002/anie.201206457>; <https://doi.org/10.1021/ja903157e>). So, I also suggest that the authors delete the phrase “could be a new approach to regulate molecular aggregation and morphology characteristics” instead of citing the corresponding references in related research fields.

Response: We thank the reviewer for the insightful comments and suggestions. In solution, the additional absorption band at 689 nm can be attributed to the formation of H-aggregates of the DTC8 moiety. Moreover, we fully agree with the reviewer that the significant blue shift observed for Six-IC may result from its dendritic structure, which effectively prevents molecular aggregation. As suggested, the recommended references (references 54 and 55) have been incorporated into the revised main text and highlighted. Additionally, the phrase “supporting our hypothesis that the designed molecular configuration for GMAs could be a new approach to regulate molecular aggregation and morphology characteristics” has been removed from the revised main text.

“which is mainly attributed to the H-aggregates of the DTC8 moiety.”

Main text, Page 6

“and dendritic structure can effectively regulates molecular aggregation.”

Main text, Page 23

50. Lo S-C, Harding RE, Shipley CP, Stevenson SG, Burn PL, Samuel IDW. High-Triplet-Energy Dendrons: Enhancing the Luminescence of Deep Blue Phosphorescent Iridium(III) Complexes. J. Am. Chem. Soc. 131, 16681-16688 (2009).

51. Tang M-C, Tsang DP-K, Chan MM-Y, Wong KM-C, Yam VW-W. Dendritic Luminescent Gold(III) Complexes for Highly Efficient Solution-Processable Organic Light-Emitting Devices. Angew. Chem. Int. Ed. 52, 446-449 (2013).

7. As stated by the authors in the article, “This indicates that Six-IC and DTC8 follow different aggregation motifs in liquid and solid states, supporting our hypothesis that the designed molecular configuration for GMAs could be a new approach to regulate molecular aggregation and morphology characteristics.” Could the authors provide temperature-dependent absorption spectra of Six-IC and DTC8 in dilute solution to further explore their distinct aggregation behaviors?

Response: We thank the reviewer for the insightful comments and suggestions. Following the recommendation, we conducted temperature-dependent absorption measurements for Six-IC and DTC8 in both dilute CF and CB solutions. As shown in **Fig. R3**, with increasing temperature, DTC8 exhibits slightly larger blue shifts compared to Six-IC. Notably, Six-IC displays H-aggregate peaks around 690 nm in both CF and CB solutions, and these H-aggregates remain relatively stable as the temperature increases, further confirming that the dendritic molecular structure is responsible for the formation of H-aggregates.

Fig. R3. Temperature-dependent absorption spectra of DTC8 (a, c) and Six-IC (b, d) in dilute CF and CB solutions.

8. The data in Supplementary Table 1 need to be consistent with the main text. The authors should double-check and correct this discrepancy.

Response: We thank the reviewer for their insightful comments and suggestions. The data presented in **Supplementary Table 1** were preliminary results. After careful optimization to improve the J_{sc} and FF parameters, the efficiency was increased to 19.4%. Following the reviewer's recommendation, we have updated the data in **Supplementary Table 1** to ensure consistency with the main text.

9. According to the GIWAXS patterns, Six-IC exhibits a less oriented molecular packing motif, which is generally considered unfavorable for charge transport. How do the authors address this issue?

Response: We thank the reviewer for highlighting this interesting issue. Charge mobility is indeed influenced by a wide range of factors, extending beyond crystalline

and molecular packing features. These factors include, but are not limited to, intermolecular interactions (*Adv. Mater.* **13**, 1053-1067 (2001)); *J. Mater. Chem.* **22**, 20840-20851 (2012); *Proc. Natl. Acad. Sci.* **112**, 10599-10604 (2015); *Chem. Mater.* **32**, 7338–7346 (2020)), intramolecular charge transfer (*J. Am. Chem. Soc.* **127**, 16866–16881 (2015); *Sol. Energy Mat. Sol. C* **123**, 112-121 (2014)), domain continuity (*Adv. Mater.* **22**, 3812-3838 (2010); *Chem. Commun.* **48**, 5859-5861 (2012); *Adv. Energy Mater.* **7**, 1700888 (2017)), etc. In this context, the CVT-tethered dendritic structure of Six-IC facilitates enhanced interaction between the SMA arms, promoting more efficient charge transport. Consequently, Six-IC exhibits a higher charge mobility ($1.49 \times 10^{-3} \text{ cm}^2 \text{ V}^{-2} \text{ s}^{-1}$) than DTC8 ($8.31 \times 10^{-4} \text{ cm}^2 \text{ V}^{-2} \text{ s}^{-1}$) as shown in **Supplementary Fig. 5**. Although the mobilities decrease due to morphology changes after blending, the values for **Six-IC** in blend films still remain higher than those of **DTC8**. Following the suggestion, **Supplementary Fig. 5** has been added to the revised **Supplementary Information** and highlighted accordingly.

Supplementary Fig. 5. SCLC plots of the (a) hole-only and (b) electron-only devices. The experimental data are fitted using the SCLC model (solid lines).

10. The PiFM results are intriguing and informative. However, additional measurement details should be provided, such as the characteristic FT-IR peaks of D18, DTC8, and Six-IC.

Response: We thank the reviewer for the insightful comments and suggestions. In line with this recommendation, additional measurement details, including the characteristic FT-IR peaks of D18, DTC8, and Six-IC, have been incorporated into the revised **Supplementary Fig. 13** in Supplementary Information and highlighted accordingly.

Supplementary Information, Page 16

14. Photoinduced Force Microscopy (PiFM) Measurement

Nanoscale chemical imaging and infrared spectroscopy were conducted using an AFM-IR nanoIR3 system (Bruker, USA) equipped with a quantum cascade laser (QCL). For nanoscale chemical imaging, the amplitude corresponding to specific laser wavelengths was utilized. Nanoscale IR spectra were obtained by measuring the amplitude of cantilever oscillations within the range of 800 to 1800 cm^{-1} , with a resolution of 2 cm^{-1} . This was achieved using a gold-coated silicon cantilever featuring a nominal spring constant of 0.3 N m^{-1} .

Supplementary Fig. 13. The characteristic FT-IR peaks of D18, DTC8, and Six-IC.

11. Some sentences, including but not limited to” which is unprecedented”, should be deleted to avoid exaggeration.

Response: We thank the reviewer for the comments and suggestions. In line with this recommendation, the relatively exaggerated statements, such as "which is unprecedented" (Page 4) and "an unprecedented report" (Page 17), have been removed from the revised main text.

12. On page 8, during the author's discussion of the DFT calculation, they mentioned that the two SMA units are considered for use in the calculation. The reason for that is time-saving. I suggest the author note that the electronic structure is not altered by SMA units, and the simulation is not affected by reducing the molecular complexity, which should be the better reason for this compromise between calculation time and simulation accuracy. Please edit accordingly.

Response: We thank the reviewer for the comments and fully agree with the reviewer’s recommendation. In response, the original statement, “To save time, only two SMA subunits on Six-IC was used for calculated,” has been revised to: “Considering that the electronic structure is not significantly affected by the number of SMA units, and to balance calculation time with simulation accuracy, a simplified Six-IC model containing only two SMA subunits was used for the calculations.” This revision has been incorporated and highlighted in the main text.

Reviewer #2 (Remarks to the Author):

The manuscript reports a new macromolecular hexamer acceptor with a branch-connecting strategy, resulting in a molecular weight exceeding 10 KDa. The binary OSCs based on D18 achieve an efficiency of 19.4%. While the study presents a new case for the use of high-molecular-weight acceptors in OSCs, a more detailed mechanistic understanding of the structure-property relationship and the advantages of this design, would enhance its quality. Also, several critical issues should be addressed for further investigation:

Response: We thank the reviewer for the positive comments and recommendation for publication of this work.

1. The authors state in the MD simulations, “To save time, only two SMA subunits on Six-IC were used for the calculations.” However, for a high-molecular-weight acceptor with the cup-shaped conformation of cyclotrimeratrylene, the number of SMA subunits is crucial for understanding the intramolecular packing behavior. As a result, the current DFT calculations lack scientific significance and cannot serve as a reliable reference.

Response: We thank the reviewer for their insightful comments and suggestions. In this study, DFT calculations were primarily employed to optimize molecular structures and analyze the electrostatic potential (ESP), HOMO and LUMO energy levels, and electron and hole distributions. Given the non-conjugated connection between the CVT core and the SMA arms, the electronic structure is not significantly influenced by the number of SMA units. To balance computational feasibility with simulation accuracy, a simplified Six-IC model containing only two SMA subunits was used for these calculations. Additionally, to determine packing distances, we used DFT calculations based on this simplified model, as performing such calculations on the full Six-IC molecule would be computationally prohibitive. To supplement this limitation, molecular dynamics (MD) simulations (shown in **Fig. 2d** of the main text) were also conducted to gain insights into the intramolecular packing of the entire Six-IC molecule. We also acknowledge and agree with the reviewer’s observation that the current simplified Six-IC model lacks sufficient scientific rigor for investigating intramolecular packing behavior. Accordingly, the discussion on packing distances has been revised. Specifically, the original text—“Due to the cup-shaped conformation of cyclotrimeratrylene, the SMA subunits of Six-IC remain on the same side. In the DTC8 model, the π – π stacking distance between two molecules was found to be 3.5 Å (**Supplementary Fig. 2b**). Meanwhile, the packing of the SMA subunits

within Six-IC is more compact (packing distance = 3.1 Å) (**Supplementary Fig. 2c**), primarily due to the combination of the branched structure and flexible spacer linkage, which allows the SMA subunits to stack without obstruction"—has been removed from the DFT calculations section. Moreover, additional discussion about intramolecular packing behavior has been incorporated into the MD simulation section. The annotated packing distances previously shown in **Supplementary Fig. 2b** and **2c** have also been removed. These updates have been highlighted in the revised manuscript.

Main text, Page 9

“Due to the cup-shaped conformation of cyclotrimeratrylene, the SMA subunits of Six-IC remain on the same side.”

Main text, Page 10

“These results were primarily attributed to the combination of the branched structure and flexible spacer linkage, which allows the SMA subunits to stack without obstruction.”

2. The significant blue-shift observed in the Six-IC films, approximately 46 nm relative to DTC8, typically suggests an elevation in the E_{LUMO} energy level. How, then, can the authors explain the nearly identical E_{LUMO} levels of Six-IC (-3.92 eV) and DTC8 (-3.91 eV)? Also the blue shift in the absorption should be explained for the new acceptor.

Response: We thank the reviewer for the comments and suggestions. We agree the reviewer that the blue shifts in absorption is usually associated with a change in energy levels. To clarify this issue and also following the **Reviewer #3**'s suggestion, we have measured the UPS and IPES to recalculate the energy levels. As shown in **Supplementary Fig. 4**, the HOMO/LUMO energy levels obtained by UPS/IPES were

-5.31 /-2.93 eV for D18, -5.89 /-4.05 eV for DTC8, and -5.84 /-3.92 eV for Six-IC, respectively. These findings indicate a simultaneous increase in the HOMO and LUMO energy levels for the dendritic molecule Six-IC. Furthermore, due to significant discrepancies between energy levels measured by CV measurements and those by UPS/IPES, we have replaced **Supplementary Fig. 4** with UPS/IPES figures as suggested. The UPS/IPES-based energy levels have been updated in **Fig. 1d**, **Table 1**, and throughout the main text. Measurement details have been added to the general characterizations section, and the discussion on energy levels in the main text has been rephrased and highlighted accordingly. In page 8, the sentence “Six-IC received similar HOMO and LUMO energy levels with DTC8.” was deleted in main text.

Supplementary Fig. 4. The derived LUMO/HOMO energy levels by (a-c) IPES and (d-f) UPS measurements.

Main text, Page 7

“The HOMO/LUMO energy levels obtained by UPS/IPES were -5.89/-4.05 eV for DTC8, and -5.84/-3.92 eV for Six-IC, respectively. These results indicate a simultaneous increase in both the HOMO and LUMO energy levels for the dendritic molecule Six-IC. The elevated LUMO energy level of Six-IC is favorable for achieving

a higher open-circuit voltage.”

Main text, Page 20

“Ultraviolet photoelectron spectroscopy (UPS) measurements were recorded using a RIKEN KEIKI spectrometer (Model AC-3). Inverse photoemission spectroscopy (IPES) measurement was performed using a customized ULVAC-PHI LEIPS instrument with Bremsstrahlung isochromatic mode.”

Supplementary Information, Page 6

“The ITO samples coated with ~50 nm films of D18, DTC8, and Six-IC were used for IPES measurements.”

3. It is widely accepted that the V_{oc} values of OSC devices are generally positively correlated with the difference between the E_{HOMO} of the donor and the E_{LUMO} of the acceptor. However, the significant difference in V_{oc} values for Six-IC and DTC8-based devices, despite their similar E_{LUMO} values, is not sufficiently explained by the energy loss analysis presented in Figure 3h.

Response: We thank the reviewer for the comments and suggestions. We agree the reviewer that the V_{oc} values of OSC devices are generally positively correlated with the difference between the E_{HOMO} of the donor and the E_{LUMO} of the acceptor. As the response to the **Question 2** and **Reviewer #3's** concern, we have measured the UPS and IPES, which is considered a more convincing method to recalculate energy levels. As shown in **Supplementary Fig. 4**, the HOMO/LUMO energy levels obtained by UPS/IPES were -5.31 /-2.93 eV for D18, -5.89 /-4.05 eV for DTC8, and -5.84 /-3.92 eV for Six-IC, respectively. These findings indicate a simultaneous increase in the HOMO and LUMO energy levels for the dendritic molecule Six-IC and the significant difference in V_{oc} values was mainly attributed to the upward shift of LUMO level. Furthermore, we have replaced **Supplementary Fig. 4** with UPS/IPES figures as suggested. The UPS/IPES-based energy levels have been updated in **Fig. 1d**, **Table 1**, and throughout the main text. Measurement details have been added to the general characterizations section, and the discussion on energy levels in the main text

has been rephrased and highlighted accordingly. In page 8, the sentence “Six-IC received similar HOMO and LUMO energy levels with DTC8.” was deleted in main text.

Supplementary Fig. 4. The derived LUMO/HOMO energy levels by (a-c) IPES and (d-f) UPS measurements.

Main text, Page 7

“The HOMO/LUMO energy levels obtained by UPS/IPES were -5.89/-4.05 eV for DTC8, and -5.84/-3.92 eV for Six-IC, respectively. These results indicate a simultaneous increase in both the HOMO and LUMO energy levels for the dendritic molecule Six-IC. The elevated LUMO energy level of Six-IC is favorable for achieving a higher open-circuit voltage.”

Main text, Page 20

“Ultraviolet photoelectron spectroscopy (UPS) measurements were recorded using a RIKEN KEIKI spectrometer (Model AC-3). Inverse photoemission spectroscopy (IPES) measurement was performed using a customized ULVAC-PHI LEIPS instrument with Bremsstrahlung isochromatic mode.”

Supplementary Information, Page 6

“The ITO samples coated with ~50 nm films of D18, DTC8, and Six-IC were used for IPES measurements.”

4. In the fs-TAS experiments, the authors state that “D18 contains a faster hole transfer, thus leading to quicker exciton dissociation toward boosted J_{sc} ” and that “a longer charge lifetime and less recombination occur in D18.” However, the observed lower J_{sc} for D18 devices compared to D18 challenges this conclusion and necessitates a more compelling explanation.

Response: We thank the reviewer for the comment. In the main text, we have stated that "D18:Six-IC contains a faster hole transfer, thus leading to quicker exciton dissociation toward boosted J_{sc} " and "a longer charge lifetime and less recombination occur in D18." However, despite these advantages, the device shows a lower J_{sc} . The reason behind this is that Six-IC, with its dendritic structure, exhibits a higher LUMO energy level, resulting in a larger bandgap. Although D18:Six-IC devices demonstrate efficient exciton dissociation, longer charge lifetime, and reduced recombination loss, these benefits do not fully compensate for the impact of the increased bandgap, leading to a lower J_{sc} compared to D18:DTC8. Fortunately, the dendritic structure of Six-IC raises the LUMO energy level, enhancing the V_{oc} , and regulates crystallization to improve the FF, effectively compensating for the adverse effects of the bandgap change. In line with the reviewer's suggestion for clarity, the phrase "toward boosted J_{sc} " has been removed from the revised main text.

5. During device fabrication, the blend films typically exist in a thermodynamically metastable state to achieve higher efficiency, resulting in a stability curve that usually exhibits a “burn-in” loss stage. Interestingly, both Six-IC and DTC8-based devices show no such “burn-in” loss stages. This represents a significant advancement in organic photovoltaics by eliminating the “burn-in” loss, which should be highlighted and well explained in the abstract. If the case, this can be compared with other results ?

Additionally, the manuscript should provide detailed information regarding the stability testing methodology, including the model of the stability characterization equipment used.

Response: We would appreciate the reviewer for his/her acute observation. We completely agree with the reviewer that the morphological metastable state during the initial stage can lead to burn-in loss of the device. However, both Six-IC- and DTC8-based devices exhibit no such "burn-in" loss, because the devices were pre-aged for 72 hours prior to stability testing for better comparing the effect of morphological evolution on long term stability and if possible, extracting the T_{80} life. As one of the most widely used metric to describe the operational lifetime of a device or module, T_{80} life was defined as the period of time that elapses between the initial stabilized performance and the point where 80% of the initial performance has been reached. Since the device performance may be decreasing or increasing before initial stabilization, the estimation of T_{80} often omit any "burn in" period (*Adv. Energy Mater.* **1**, 491–494 (2011); *Sol. Energy Mat. Sol. C.* **95**, 1253–1267 (2011); *Joule* **2**, 1019–1027(2018); *Sci. Bull.* **65**, 208-216 (2020)). To maintain academic rigor, we have incorporated the detailed stability testing methodology in the **Supplementary Information** and highlighted.

Supplementary Information, Page 17

"16. Stability Measurement

The testing devices were fabricated under the same preparation conditions as those used for the J-V curve measurements. These samples were then transferred to a nitrogen-filled glovebox, maintaining the same controlled water-oxygen environment. The devices were placed on a thermal-conducting copper plate, with insulating glass pieces inserted between the copper plate and the device. The temperature of the copper plate was controlled to either room temperature or 85°C via heat conduction through the glass sheet, providing a stable thermal environment for the device

measurements. Before the data collection, the devices was pre-aged for 72 hours in the room temperature or 85°C. The J-V characteristics of the devices were regularly checked, and the photovoltaic parameters were automatically calculated based on the resulting J-V curves.”

6. The figures should standardize line thickness, font size, and formatting to ensure consistency and clarity.

Response: We thank the reviewer for the comments and suggestions. Following this suggestion, the format of all figures has been standardized.

7. The significant progress in OPV, particularly with ITIC and its derivatives, should be supported by relevant references that highlight not only advancements in efficiency but also important milestone works, as well as the mechanistic studies.

Response: We thank the reviewer for the comments and suggestions. Following this suggestion, the references for milestone works (ITIC, Y6) and their relevant mechanistic studies were added in the reference section and highlighted.

Main text, Page 22

17. *Lin Y, et al. An electron acceptor challenging fullerenes for efficient polymer solar cells. Adv. Mater. 27, 1170-1174 (2015).*
18. *Yuan J, et al. Single-Junction Organic Solar Cell with over 15% Efficiency Using Fused-Ring Acceptor with Electron-Deficient Core. Joule 3, 1140-1151 (2019).*
19. *Wang J, Xue P, Jiang Y, Huo Y, Zhan X. The principles, design and applications of fused-ring electron acceptors. Nat. Rev. Chem. 6, 614-634 (2022).*
20. *Yi J, Zhang G, Yu H, Yan H. Advantages, challenges and molecular design of different material types used in organic solar cells. Nat. Rev. Mater. 9, 46-62 (2024).*

8. The synthesis of supermolecules remains challenging, yet the yields (over 80-90%) achieved in this work are significantly higher than those in previous reports, even with a molecular weight exceeding 10 kDa. This should be thoroughly explained for readers, supported with detailed comparisons to prior studies, and highlighted in the manuscript.

Response: We thank the reviewer for the comments and suggestions. We agree that the overall yield of 58% for Six-IC is highly impressive and stands as one of the key highlights of this work. The high yield can be attributed to the following key factors: **(1) Efficient Synthesis of Six-H:** Although the synthesis of the intermediate Six-H involves the connection of six arms, the nucleophilic reaction between brominated alkanes and phenolic hydroxyl groups is straightforward and typically proceeds smoothly with yields >95% (*Tetrahedron Lett.*, **19**, 4581 (1979); *Tetrahedron* **36**, 1223 (1980); *Chemistry Select* **2**, 4892 (2017); *Molecules* **2**, 602 (2020)). This ensures a high yield for Six-H (**Fig. R1**). **(2) High-Yield Reactions:** The conversion of Six-H to Six-CHO and from Six-CHO to Six-IC is accomplished using well-established, high-yield reactions: the Vilsmeier-Haack and Knoevenagel condensation reactions. These are commonly used in semiconductor material synthesis, typically achieving yields >90% (*Green Chem.* **10**, 873 (2008); *Green Chem.* **10**, 767 (2008); *Inorg. Chem.* **59**, 15, 10578 (2020); *Dalton Trans.* 50, 4445 (2021); *J. Am. Chem. Soc.* **144**, 8, 3653 (2022)). **(3) Branching Linkage Strategy:** The branching linkage strategy ensures that the chemical reactivity of each arm remains independent, which contributes to the efficient reactions and high yield of the final product (**Fig. R1**). The use of such strategies for designing multi-armed acceptors has been shown to yield target products consistently in high yields (~90%) (*Nat. Commun.* **14**, 2926 (2023); *Angew. Chem. Int. Ed.* **63**, e202400590 (2024)). Following the reviewer's suggestion, we have added the overall yields of representative GMAs to **Supplementary Table 1**, and the corresponding discussion has been incorporated into the revised main text to emphasize the high yield metric. Additionally, to ensure better reproducibility, we have provided a more detailed synthesis process for the key intermediate Six-H in the

Supplementary Information, with the updates highlighted.

Fig. R1. The chemical structures of Com 2, IC, Six-H, Six-CHO and Six-H, along with the TLC retention factors.

Main text, Page 4-5

“The effectiveness of the nucleophilic reaction between brominated alkanes and phenolic hydroxyl groups, combined with the non-conjugated CVT core, ensures that the chemical reactivity of each arm remains independent. These factors collectively contribute to the high yield of the key intermediate, Six-H.”

Main text, Page 5

“This value represents a relatively high yield for synthesizing high-molecular-weight GMAs (Supplementary Table 1), underscoring the potential of the branched molecule design strategy for acceptor synthesis and future applications.”

9. For the DSC curves of Six-IC, to check the result of T_C values, different scan rates should be proved. The melting enthalpy (ΔH_m) of Six-IC significantly decreased from 23.85 J/g to 9.07 J/g. How can this change be explained effectively? This should be proved the manuscript. The authors states “This result was further confirmed by DSC. How can this be explained?”

(1) For the DSC curves of Six-IC, to check the result of T_c values, different scan rates should be proved.

Response: We sincerely thank the reviewer for their careful observations. The baseline of Six-IC's DSC curve does indeed appear uneven (**Fig. R2**), which could significantly impact the estimation of the T_c value. Following the reviewer's suggestion, we remeasured the DSC at heating and cooling rates of $5^\circ\text{C}/\text{min}$ and $10^\circ\text{C}/\text{min}$, respectively. As shown in **Fig. R4**, we did not observe the cold crystallization temperature around 170°C for Six-IC at either scan rate. Furthermore, the two DSC curves showed similar ΔH_{ms} values (10.8 J/g for the $5^\circ\text{C}/\text{min}$ rate and 9.47 J/g for the $10^\circ\text{C}/\text{min}$ rate), with the same T_m temperature of 254.4°C for both scan rates. This T_m temperature is also consistent with the value (255°C) reported in the main text. These results indicate that the dendritic structure of Six-IC effectively eliminates the cold crystallization behavior, likely due to the restricted molecular motion. Therefore, to provide more precise data, we have updated the DSC curves of Six-IC at the $10^\circ\text{C}/\text{min}$ scan rate, replacing the data presented in **Fig. 1g**. Additionally, the discussion regarding the DSC data has been revised and updated in the main text.

Fig. R2. DSC curves of DTC8 and Six-IC measured at a heating or cooling rate of $10^\circ\text{C}/\text{min}$.

Fig. R4. DSC curves of Six-IC measured at heating and cooling rates of 5 °C/min and 10 °C/min.

Main text, Page 7

“DTC8 exhibited a cold crystallization temperature (T_c) of 150 °C, while Six-IC showed no noticeable T_c . Additionally, Six-IC displayed a much higher melting temperature (T_m) at 254 °C compared to DTC8, which melted at 203 °C. The melting enthalpy (ΔH_m) of Six-IC decreased significantly, from 23.85 J/g to 9.47 J/g.”

(2) The melting enthalpy (ΔH_m) of Six-IC significantly decreased from 23.85 J/g to 9.07 J/g. How can this change be explained effectively?

Response: We thank the reviewer for their comments. The disappearance of cold crystallization behavior and the significant decrease in ΔH_m can be primarily attributed to the suppressed crystallization and the increased intermolecular interactions, which restrict molecular motion. These phenomena have been commonly reported in studies of giant molecule acceptors (*Nat. Energy* 7, 1180-1190 (2022); *Adv. Mater.* 35, 2206563 (2023); *Adv. Mater.* 36, 2308606 (2024)). In fact, we have already provided a description in the main text: “These findings suggest that the newly developed branched giant acceptors can effectively suppress crystallization, potentially limiting the diffusion of SMA in polymer blends.⁵⁷” This statement sufficiently describes the reasons behind the distinct ΔH_m observed in our study.

(3) The authors states “This result was further confirmed by DSC. How can this be explained?”

Response: We thank the reviewer for their helpful comments. Our intended meaning for this sentence was to convey that a higher T_g , as estimated from temperature-dependent film absorption, typically correlates with suppressed molecular movement, contributing to a more stable morphology. Similarly, a higher cold crystallization temperature or melting temperature, along with a lower melting enthalpy in the DSC test, generally indicates suppressed crystallization, which also promotes stable morphology. Following the reviewer’s suggestion and recognizing the fundamentally different principles of the UV-vis deviation metric (DMT) method and DSC measurements, we have deleted the sentence “This result was further confirmed by DSC” to ensure greater rigor in the revised manuscript.

10. The authors state “the enhanced miscibility resulting from increased donor/acceptor interface contact” from GIWAXS, how can this be explained?

Response: We thank the reviewer for their comments and apologize for the imprecise expression in the original manuscript. Our original intention was to explain that the slightly reduced 200 peak intensity in the in-plane (IP) direction is primarily attributed to enhanced miscibility, which can increase donor/acceptor interface contact. To address this, we have revised the text to remove the phrase “resulting from increased donor/acceptor interface contact” and updated the discussion in the revised main text for improved clarity and precision.

11. In Figure 6, the evidence for morphology stability is provided. However, is there any significant difference between the morphologies of the two blends based on this data? This should be discussed with the PCEs.

(1) In Figure 6, the evidence for morphology stability is provided. However, is there any significant difference between the morphologies of the two blends based on this data?

Response: We thank the reviewer for this comment. Indeed, significant differences in morphological changes were observed when the blend films were heated at 150 °C for 3 hours. As shown in **Fig. 4a** and **Fig. 6e**, after heating, the RMS values of the D18:DTC8 films increased significantly from 1.41 nm to 9.17 nm. Correspondingly, the TEM images revealed a transition from a uniform interpenetrating network structure (**Fig. 4c**) to large-scale phase separation (**Fig. 6f**). In contrast, when annealed at 150 °C for 3 hours, the RMS values of the D18:Six-IC films showed minimal change, increasing only slightly from 1.04 nm (**Fig. 4b**) to 1.56 nm (**Fig. 6g**). A similar trend was observed in the TEM measurements, where the D18:Six-IC film maintained an interpenetrating network structure with well-defined phase domains both before and after heating. In response to the comment, the RMS values for **Fig. 4a**, **4b**, **6e**, and **6g** have been labeled in the respective figures in the revised main text. The corresponding discussion has also been added to the main text and highlighted for clarity.

Main text, Page 17

*“When annealed at 150 °C for 3 hours, the D18:Six-IC films exhibited minimal changes, whereas the D18:DTC8 films showed a significant increase in RMS values from 1.41 nm (**Fig. 4a**) to 9.17 nm (**Fig. 6e**) in the AFM images, along with large-scale phase separation in the TEM images (**Fig. 4c** and **6f**).”*

(2) This should be discussed with the PCEs.

Response: In response to the reviewer’s suggestion, the *J-V* curves of the devices before and after heating at 150 °C are now provided in the Supplementary Information. As shown in **Supplementary Fig. 14**, after being heated at 150 °C for 3 hours, the device efficiency of DTC8 decreased significantly from 17.2% to 13.1%,

whereas the efficiency of Six-IC showed only a slight decline from 19.3% to 18.1%. This result clearly demonstrates the effectiveness of the dendritic acceptor strategy in enhancing the morphological stability of the films and the thermal stability of the corresponding devices. Following this suggestion, the following discussion was added in the revised main text and highlighted:

Main text, Page 17

“Meanwhile, the OSC device based on D18:Six-IC retained 93.8% of its initial efficiency, whereas the one based on D18:DTC8 retained only 76.2%, (Supplementary Fig. 14)”

12. D18:Six-IC blend films shows high mobility while its crystallization is greatly suppression. How can this be explained?

Response: We thank the reviewer for the comments. We would appreciate the reviewer’s focusing on this interesting issue. Comprehensively speaking, charge mobility is determined by a lot of factors including but beyond crystalline and molecular packing features, e.g.: intermolecular interaction (*Adv. Mater.* **13**, 1053-1067 (2001)); *J. Mater. Chem.* **22**, 20840-20851 (2012); *Proc. Natl. Acad. Sci.* **112**, 10599-10604 (2015); *Chem. Mater.* **32**, 7338–7346 (2020)), intramolecular charge transfer (*J. Am. Chem. Soc.* **127**, 16866–16881 (2015); *Sol. Energy Mater. Sol. Cells* **123**, 112-121 (2014)), domain continuity (*Adv. Mater.* **22**, 3812-3838 (2010); *Chem. Commun.* **48**, 5859-5861 (2012); *Adv. Energy Mater.* **7**, 1700888 (2017)), etc. In this context, the CVT-tethered dendritic structure enhances the interactions between the SMA arms, facilitating improved charge transport within the Six-IC. Consequently, Six-IC exhibits higher charge mobility ($1.49 \times 10^{-3} \text{ cm}^2 \text{ V}^{-2} \text{ s}^{-1}$) than DTC8 ($8.31 \times 10^{-4} \text{ cm}^2 \text{ V}^{-2} \text{ s}^{-1}$) (**Supplementary Fig. 5**) despite the suppressed crystallization. This higher charge mobility is effectively maintained in the corresponding blend films, therefore ensuring higher mobility values for Six-IC blend

films.

(13) The ^{13}C of all the new materials should be provided. also the TGA.

Response: We thank the reviewer for the comments and suggestions. Following these recommendations, we have added the related ^{13}C NMR, TGA, and also high-temperature GPC data in the revised **Supplementary Information**. The ^1H NMR, ^{13}C NMR, mass spectra, and GPC data collectively confirm the successful synthesis of the intermediates and final products. In the TGA curves (**Supplementary Fig. 1**), the intermediates Six-H and Six-CHO exhibit 5% decomposition temperatures above 370 °C, indicating the good stability of the dendritic core structure. Upon attachment of the IC terminal, Six-IC shows a decomposition temperature of 332 °C, which is slightly lower than that of the intermediates but nearly identical to DTC8. These results confirm that the dendritic structure does not compromise the stability of the acceptor molecule. The ^{13}C NMR and TGA data, along with the corresponding discussion, have been added to the revised main text and Supplementary Information.

Supplementary Information, Page 2, Six-H

^{13}C NMR (151 MHz, CDCl_3) δ 147.69, 147.55, 147.51, 142.12, 142.05, 137.12, 136.92, 136.81, 136.71, 132.12, 131.45, 130.97, 123.58, 123.22, 122.82, 122.75, 119.19, 119.08, 116.11, 111.83, 111.41, 69.34, 54.73, 50.95, 38.64, 31.96, 31.95, 31.92, 30.45, 30.26, 29.73, 29.68, 29.60, 29.55, 29.50, 29.39, 29.30, 29.20, 29.05, 28.93, 28.82, 26.38, 25.83, 25.77, 25.43, 22.71, 14.14.

Supplementary Information, Page 3, Six-CHO

^{13}C NMR (151 MHz, CDCl_3) δ 181.44, 181.40, 181.30, 147.76, 147.41, 147.27, 147.18, 146.67, 146.61, 146.56, 143.12, 143.05, 142.99, 142.84, 142.47, 142.37, 137.82, 137.63, 136.92, 136.82, 136.79, 136.63, 136.55, 136.27, 135.73, 133.59, 133.07, 132.68, 132.23, 132.20, 132.07, 132.01, 129.71, 129.47, 129.19, 128.99, 127.57, 127.35, 123.40, 123.03, 122.66, 119.90, 116.22, 112.43, 112.19, 110.82, 69.42,

54.99, 54.83, 51.11, 38.97, 38.84, 36.41, 31.93, 31.92, 31.88, 30.89, 30.38, 30.35, 30.27, 29.71, 29.67, 29.64, 29.57, 29.51, 29.47, 29.36, 29.32, 29.26, 29.20, 29.11, 28.81, 28.12, 28.04, 26.57, 25.91, 25.45, 22.70, 22.67, 14.13.”

Supplementary Information, Page 4, Six-IC

¹³C NMR (126 MHz,) δ 185.49, 185.33, 157.84, 157.79, 157.55, 155.34, 155.26, 155.14, 153.84, 153.80, 153.55, 153.30, 153.16, 153.05, 148.81, 148.59, 147.39, 147.38, 147.05, 145.43, 145.40, 145.20, 137.17, 137.01, 136.62, 136.56, 135.03, 134.59, 134.45, 134.40, 134.11, 133.93, 133.23, 133.04, 132.82, 132.77, 132.62, 132.57, 131.40, 130.87, 130.80, 119.69, 114.73, 114.57, 113.75, 113.07, 112.02, 70.34, 69.23, 55.57, 51.80, 39.67, 31.98, 31.89, 31.82, 31.42, 31.35, 30.85, 30.52, 30.35, 29.93, 29.68, 29.58, 29.51, 29.38, 29.26, 29.19, 27.81, 26.94, 26.25, 26.19, 22.70, 22.62, 22.58, 14.07, 13.99, 13.96.”

Supplementary Information, Page 19

Supplementary Fig. 16. ¹³C NMR spectrum of **Six-H** in CDCl₃.

Supplementary Fig. 19. ^{13}C NMR spectrum of Six-CHO in CDCl_3 .

Supplementary Fig. 22. ^{13}C NMR spectrum of Six-IC in $\text{C}_2\text{D}_2\text{Cl}_4$ at $100\text{ }^\circ\text{C}$.

Supplementary Information, Page 23

17. Gel Permeation Chromatography Measurement

The molecular weights of Six-IC was obtained on an Acquity Advanced Polymer Chromatography (Waters) with a high-temperature chromatograph in 1,2,4-trichlorobenzene at $150\text{ }^\circ\text{C}$ and using a calibration curve of polystyrene standards.

Workbook Details

Eluent: TCB stabilised with 0.0125% BHT
Column Set: PLgel MIXED-B LS 300x7.5mm x2
Detector: RI
Flow Rate: 1.00 ml/min
Temperature: 150
Injection Volume: 200.0 ul

Analysis Using Method: 20240901

Comments:

Calibration Used: 2024/9/7 10:39:10

High Limit MW RT: 10.95 mins

High Limit MW: 6217712

K: 17.5000

Low Limit MW RT: 17.27 mins

Low Limit MW: 623

Alpha: 0.6700

MW Averages

Peak No	Mp	Mn	Mw	Mz	Mz+1	Mv	PD
1	11942	10001	10974	11844	12624	10792	1.09729

Processed Peaks

Peak No	Name	Start RT (mins)	Max RT (mins)	End RT (mins)	Pk Height (mV)	% Height	Area (mV.secs)	% Area
1		15.18	15.60	16.50	61.8949	100	1700.27	100

Supplementary Fig. 24. High-temperature GPC curve of Six-IC.

Main text, Page 6

“Thermogravimetric analysis results indicate that the dendritic structure design does not raise any stability concerns.”

Supplementary Information, Page 4-5

2. Thermogravimetric and Differential Scanning Calorimetry Analyses

Thermogravimetric (TG) measurements were conducted using a NETZSCH TG209F3 apparatus, with a heating rate of 10 °C/min under a nitrogen atmosphere. Differential scanning calorimetry (DSC) analysis was conducted using a NETZSCH DSC200F3 apparatus, with a heating and cooling rate of 10 °C/min under a nitrogen atmosphere.

Supplementary Fig. 1. Thermogravimetric analysis (TGA) curves for DTC8, Six-IC, and the key intermediates Six-H and Six-CHO.

Reviewer #3 (Remarks to the Author):

The authors present a macromolecular hexamer acceptor with a molecular weight of 10k g/mol. This hexamer addresses the molecular weight limitations of GMAs and achieves a high production yield exceeding 58%. The macromolecular acceptor, Six-IC, demonstrates enhanced crystallinity and miscibility with the donor, leading to superior morphology compared to its monomer counterpart, DTC8. The binary OSCs based on the D18 system exhibit a high efficiency of 19.4% for high-molecular-weight acceptors, alongside notable device stability and film ductility. Extensive characterization of the molecular structure and PV devices was performed to support the conclusions, supplemented by DFT and MD simulations, which are presented clearly for readers' comprehension. However, the reviewer remains concerned that numerous studies on similar GMA concepts have been published in the past three years. Despite this concern, the manuscript's structure and related characterization data are well-organized. As the authors noted, the high molecular weight and production yield achieved in this work have the potential to advance the GMA field, alongside a 19.4% efficiency and improved thermal stability. Enclosed are some suggestions for improving the manuscript before publication:

Response: We thank the reviewer for the positive comments and recommendation for publication of this work.

1). The authors list Molecular Weight vs. PCE in Figure 3c to highlight the novelty of this work. As the authors also claim that the synthesis of Six-IC is at a high yield level among GMAs, the reviewers suggest plotting a figure on the related synthesis yield. Additionally, the authors could provide some discussions on why the yield is high for such high molecular weight acceptor. This would help readers understand the rationale behind this GMA, which is related to the novelty of this work.

Response: We thank the reviewer for the comments and suggestions. The high yield of Six-IC can be attributed to the following reasons: (1) The synthesis of the intermediate Six-H involves the connection of six arms, but the nucleophilic reaction

between brominated alkanes and phenolic hydroxyl groups is straightforward and typically proceeds smoothly with yields >95% (*Tetrahedron Lett.*, **19**, 4581 (1979); *Tetrahedron* **36**, 1223 (1980); *Chemistry Select* **2**, 4892 (2017); *Molecules* **2**, 602 (2020)), ensuring a high overall yield for Six-H (**Fig. R1**); (2) The conversions from Six-H to Six-CHO and from Six-CHO to Six-IC are achieved through the Vilsmeier-Haack and Knoevenagel condensation reactions, both of which are well-established, high-yield methods commonly used in semiconductor materials synthesis, typically yielding >90% (*Green Chem.* **10**, 873 (2008); *Green Chem.* **10**, 767 (2008); *Inorg. Chem.* **59**, 15, 10578 (2020); *Dalton Trans.* 50, 4445 (2021); *J. Am. Chem. Soc.* **144**, 8, 3653 (2022)); (3) The branching linkage strategy ensures that the chemical reactivity of each arm remains independent, allowing for efficient reactions and a high yield of the final product (**Fig. R1**). The use of these reactions in designing multi-armed acceptors has been previously reported to consistently yield high yields (~90%) (*Nat. Commun.* **14**, 2926 (2023); *Angew. Chem. Int. Ed.* **63**, e202400590 (2024)). In response to the suggestion, we have included the synthesis yields of high-performance GMAs in **Supplementary Table 1**. Additionally, the following discussion regarding these yields has been added to the revised main text.

Fig. R1. The chemical structures of Com 2, IC, Six-H, Six-CHO and Six-IC, along with the TLC retention factors.

Main text, Page 4-5

“The effectiveness of the nucleophilic reaction between brominated alkanes and phenolic hydroxyl groups, combined with the non-conjugated CVT core, ensures that the chemical reactivity of each arm remains independent. These factors collectively contribute to the high yield of the key intermediate, Six-H.”

2). Figure 1g shows the DSC curves of Six-IC; however, the $T_c = 178^\circ\text{C}$ is not convincing, as the DSC curve rises after 150°C . Rechecking and remeasuring the DSC would help clarify this issue.

Response: We sincerely appreciate the reviewer for their careful observations. The rise in the DSC curve after 150°C is primarily due to an uneven baseline (**Fig. R2**), which can significantly affect the estimation of the T_c value. Following the reviewer’s suggestion, we have remeasured the DSC at heating and cooling rates of $5^\circ\text{C}/\text{min}$ and $10^\circ\text{C}/\text{min}$, respectively. As shown in **Fig. R4**, at both scan rates, we did not observe the cold crystallization temperature around 170°C for Six-IC. Additionally, the two curves showed similar ΔH_m values (10.8 J/g for the $5^\circ\text{C}/\text{min}$ rate and 9.47 J/g for the $10^\circ\text{C}/\text{min}$ rate), and identical T_m values (254.4°C for both scan rates), which were consistent with the original values in the main text. This result suggests that the dendritic structure eliminates the cold crystallization behavior, likely due to restricted molecular diffusion. To ensure greater precision, the DSC curves of Six-IC at a $10^\circ\text{C}/\text{min}$ scan rate have been used to replace the original **Fig. 1g**. The corresponding discussion in the DSC section has been updated and highlighted in the revised main text.

Main text, Page 7

“DTC8 exhibited a cold crystallization temperature (T_c) of 150°C , while Six-IC showed no noticeable T_c . Additionally, Six-IC displayed a much higher melting temperature (T_m) at 254°C compared to DTC8, which melted at 203°C . Moreover,

the melting enthalpy (ΔH_m) of Six-IC decreased significantly, from 23.85 J/g to 9.47 J/g.”

Fig. R2. The DSC curves of DTC8 and Six-IC with the heating or cooling rate of 10 °C/min.

Fig. R4. The DSC curves of Six-IC with the heating or cooling rate of 5 °C/min and 10 °C/min.

3). As the authors measured the energy levels by CV, it would be preferable to confirm the HOMO/LUMO by UPS/IPES, as CV measurements can be challenging in determining the HOMO/LUMO of these PV materials.

Response: We thank the reviewer for the comment and suggestion. We agree with the reviewer that determining HOMO/LUMO energy levels through CV measurements can be challenging. In line with the reviewer’s suggestion, we have measured the

energy levels using UPS and IPES. As shown in **Supplementary Fig. 4**, the HOMO/LUMO energy levels obtained by UPS/IPES were -5.31/-2.93 eV for D18, -5.89/-4.05 eV for DTC8, and -5.84/-3.92 eV for Six-IC, respectively. These findings indicate a simultaneous increase in both the HOMO and LUMO energy levels for the dendritic molecule Six-IC. Furthermore, due to significant discrepancies between the energy levels measured by CV and those measured by UPS/IPES, we have replaced **Supplementary Fig. 4** with UPS/IPES figures as suggested. The UPS/IPES-based energy levels have been updated in **Fig. 1d**, **Table 1**, and throughout the main text. Measurement details have been added to the general characterizations section, and the discussion on energy levels in the main text has been revised and highlighted accordingly. Specifically, the sentence “Six-IC received similar HOMO and LUMO energy levels with DTC8” has been deleted from page 8 in the revised main text.

Main text, Page 7

“The HOMO/LUMO energy levels obtained by UPS/IPES were -5.89/-4.05 eV for DTC8, and -5.84/-3.92 eV for Six-IC, respectively. These results indicate a simultaneous increase in both the HOMO and LUMO energy levels for the dendritic molecule Six-IC. The elevated LUMO energy level of Six-IC is favorable for achieving

a higher open-circuit voltage.”

Supplementary Fig. 4. The derived LUMO/HOMO energy levels by (a-c) IPES and (d-f) UPS measurements.

Main text, Page 20

“Ultraviolet photoelectron spectroscopy (UPS) measurements were recorded using a RIKEN KEIKI spectrometer (Model AC-3). Inverse photoemission spectroscopy (IPES) measurement was performed using a customized ULVAC-PHI LEIPS instrument with Bremsstrahlung isochromatic mode.”

Supplementary Information, Page 6

“The ITO samples coated with ~50 nm films of D18, DTC8, and Six-IC were used for IPES measurements.”

4). Figures 6i and 6j present the thermal stability of the devices, seemingly measured PV devices after different certain aging periods. It would be helpful to include the related sample size and error bars, along with averaged or maximum values, for

clearer representation.

Response: We thank the reviewer for the comment and suggestion. Following this suggestion, the related sample size and error bars, along with the averaged or maximum values, have been updated for Figs. 6i and 6j in the main text.

Some minor comments:

5). The unit of molecular weight is missing in the manuscript. "g/mol" should be added before 10,000, or alternatively use "kDa".

Response: We thank the reviewer for the comment and suggestion. Following this suggestion, the unit of molecular weight has been unified as g/mol in the main text and highlighted accordingly.

6). In Table 1, since Mn/Mw/PDI only applies to polymers, for an organic compound with a defined molecular weight ($M_n = M_w$, PDI = 1), using M (g/mol) would be more appropriate.

Response: We thank the reviewer for the comment and suggestion. In response, we have revised **Table 1** accordingly and highlighted the changes for clarity.

7). There are some overly promotional statements, e.g.: i) Page 4, the last sentence of the Introduction: "ultra-high molecular weight." ii) Page 17, the last sentence: "advance the entire OSC field toward commercialization." This work focuses on NFAs, and "advance the entire OSC field toward commercialization" seems overstated.

Response: We thank the reviewer for pointing out these issues. In response, we have made the necessary corrections in the main text. The statement "ultra-high molecular weight" has been corrected to "high molecular weight" and highlighted. Additionally, the phrase "advance the entire OSC field toward commercialization" has been deleted from the main text.

8). Some typos: in the caption of Fig. 2: “(c) Charge Density Differences” (all capitalized). On page 17, Materials section: "PNDI-F3N," where the number 3 is subscripted.

Response: We thank the reviewer for pointing out these mistakes. In response, we have made the necessary corrections and highlighted them in the main text.

9). Open question: Did the authors consider using fully rigid aromatic structures as the core, or was this approach abandoned due to low synthesis yield? It appears that three flexible -CH₂- linkers were used in Six-IC. A fully rigid aromatic core with six arms would likely provide greater thermal stability, which is typically preferred for such structures in the OPV field.

Response: We thank the reviewer for the comments and suggestions. In fact, we initially attempted to synthesize this acceptor molecule and successfully obtained the target product, as shown in **Fig. R5**. However, when paired with D18 and PM6, this acceptor only achieved efficiencies of 16.2% and 17.1%, respectively—significantly lower than that of current mainstream non-fullerene acceptors. Subsequently, we designed the Six-IC with a flexible CVT core, which boosted the efficiency to 19.4%. The possible reason for the efficiency improvement is that the cup-shaped conformation of the CVT core arranges the SMA units on the same side, thereby enhancing the stacking and charge transport between the SMA arms. Additionally, an important consideration in our design is that the CVT core, as a derivative of calixarene, is expected to interact supramolecularly with traditional fullerene derivatives or metal ions by tuning the arm length (*J. Am. Chem. Soc.* **116**, 10346-10347 (1994); *Chem. Eur. J.* **6**, 3501-3507 (2000); *Chem. Mater.* **17**, 2063–2068 (2005); *Inorg. Chem.* **55**, 9230-9239 (2016)). These interactions, if confirmed, could open up intriguing potential applications for these dendritic acceptors in both organic and perovskite solar cells. Currently, our lab is investigating the design and synthesis of dendritic acceptors with larger molecular weights and multifunctional

features.

Fig. R5. (a) The chemical structure of fully aromatic core-based dendritic acceptor AMA; The $J-V$ curves of optimal D18:AMA (b) and PM6:AMA (c) devices.